# Dual-discriminative Graph Neural Network for Imbalanced Graph-level Anomaly Detection

**Ge Zhang**[1]    **Zhenyu Yang**[1]
**Jia Wu**[1][*]    **Jian Yang**[1]    **Xue Shan**[2]    **Hao Peng**[3]    **Jianlin Su**[4]
**Chuan Zhou**[5]    **Quan Z. Sheng**[1]    **Leman Akoglu**[6]    **Charu C. Aggarwal**[7]
[1]Macquarie University    [2]University of Wollongong    [3]Beihang University
[4]Zhuiyi Technology    [5]AMSS, Chinese Academy of Sciences
[6]Carnegie Mellon University    [7]IBM T. J. Watson Research Center
{ge.zhang5, zhenyu.yang3}@students.mq.edu.au,
{jia.wu, jian.yang}@mq.edu.au, sxue@uow.edu.au,
penghao@buaa.edu.cn, bojonesu@wezhuiyi.com, zhouchuan@amss.ac.cn,
michael.sheng@mq.edu.au, lakoglu@andrew.cmu.edu, charu@us.ibm.com

## Abstract

Graph-level anomaly detection aims to distinguish anomalous graphs in a graph dataset from normal graphs. Anomalous graphs represent a very few but essential patterns in the real world. The anomalous property of a graph may be referable to its anomalous attributes of particular nodes and anomalous substructures that refer to a subset of nodes and edges in the graph. In addition, due to the imbalance nature of anomaly problem, anomalous information will be diluted by normal graphs with overwhelming quantities. Various anomaly notions in the attributes and/or substructures and the imbalance nature together make detecting anomalous graphs a non-trivial task. In this paper, we propose a graph neural network for graph-level anomaly detection, namely iGAD. Specifically, an anomalous graph attribute-aware graph convolution and an anomalous graph substructure-aware deep Random Walk Kernel (deep RWK) are welded into a graph neural network to achieve the dual-discriminative ability on anomalous attributes and substructures. Deep RWK in iGAD makes up for the deficiency of graph convolution in distinguishing structural information caused by the simple neighborhood aggregation mechanism. Further, we propose a Point Mutual Information (PMI)-based loss function to target the problems caused by imbalance distributions. PMI-based loss function enables iGAD to capture essential correlation between input graphs and their anomalous/normal properties. We evaluate iGAD on four real-world graph datasets. Extensive experiments demonstrate the superiority of iGAD on the graph-level anomaly detection task.

## 1   Introduction

Graphs have been used as an indispensable tool to describe relational data, such as chemical compounds, transport networks, and social networks. For instance, in a graph that models a chemical compound, nodes represent atoms and edges describe chemical bonds between atoms. Learning graphs to understand the world has been drawing considerable attention in recent years [5, 14, 15, 51]. Given a graph dataset containing a set of graphs, graph-level anomaly detection aims to identify graphs with anomalous information. Generally, these anomalous graphs with special characteristics or functions are the minority in a graph dataset. For example, among all antibody molecules, only

---

[*]Corresponding Author

36th Conference on Neural Information Processing Systems (NeurIPS 2022).

a few of them exhibit anti-cancer activity on a specific type of cancer cell line [55]. Graph-level anomaly detection is a research topic of great practical value, say, it can support cancer drug discovery by identifying molecules with anti-cancer activity from other antibody molecules [48]. Also, the research can help to reveal pathogenic brain mechanisms by distinguishing the brain structures of patients with neurological disorders from healthy individuals [21].

In practice, the attributes of particular nodes are a contributing factor to a graph's anomalous properties. For example, the hydrogen and oxygen atoms at the end of molecules are the key to discriminating between Alcohol and Alkane [45]. Graph substructures can be another factor for a graph's anomalous properties. For instance, pharmacophores composed of some particular substructures can induce molecular toxicity. These graph attributes and substructures that are essential for distinguishing anomalous graphs from normal graphs are regarded as *anomalous attributes* and *anomalous substructures*, respectively. Most existing graph mining methods, especially those graph convolution-based models which have achieved state-of-the-art performance [54, 59, 20, 3], focus on the aggregation of neighborhood-structured information. However, the anomalous structure information could be all possible substructures. In addition, anomalous attributes and substructures will be diluted by the information of normal graphs with overwhelming quantities, which is the *imbalance nature* of the anomaly problem. In summary, various anomaly notions in attributes and/or substructures and the imbalance nature make graph-level anomaly detection an important yet challenging problem.

In this paper, we propose a dual-discriminative graph neural network for imbalanced Graph-level Anomaly Detection (iGAD). Specifically, iGAD investigates a novel anomalous attribute-aware graph convolution. The final node representation is the concatenation of the independent encoding on a node itself and the independent encoding on its neighborhoods from different hops. In this way, the attributes of center nodes can evolve independently over the graph convolution process and the anomalous attributes on particular nodes can get exposure. However, it is intractable to enable graph convolution to be aware of anomalous substructures. Due to the high dependence on neighborhood aggregation, graph convolution can only capture very limited substructures in graphs. Moreover, even the simple substructures such as triangles, graph convolution cannot count them in a graph [3]. RWK [47], as one type of Graph Kernels (GKs), learns graphs by exploring common random walk sequences among all input graphs, which naturally involves the comparison about diverse substructures in different graphs. To embed the information of anomalous substructures into graph representations, we propose to compare random walk sequences of input graphs against anomalous substructures. By modeling anomalous substructures as parameters, they became learnable and can get updated through back-propagation. We entitle this design anomalous substructure-aware deep RWK. In addition, we propose a new loss function to target the imbalance distribution caused by the great difference in the number of normal and anomalous graphs. The newly proposed loss function models the PMI between graphs and their normal/anomalous properties and enables iGAD to capture anomalous information from an imbalanced graph dataset. Welding three designs mentioned above into a graph neural network, the graph-level anomaly detection model iGAD is achieved. We conclude the contributions of iGAD as follows:

- The anomalous attribute-aware graph convolution exposes anomalous attributes by keeping the independent evolution of center node's attributes.

- Anomalous substructures are modeled as learnable parameters. Substructure-aware deep RWK embeds the information of anomalous substructures into graph representations by comparing the random walk sequences from input graphs and anomalous substructures.

- The PMI-based loss function addresses issues brought by the imbalance nature of the anomaly problem and enables iGAD to learn anomalous graphs as a minority effectively.

- Comprehensive experiments demonstrate the effectiveness of iGAD, which outperforms state-of-the-art graph convolution-based graph classification models and graph-level anomaly detection algorithms on four real-world graph datasets. The code is available at https://github.com/graph-level-anomalies/iGAD.

## 2 Related Work

**Anomaly Detection.** The task aims to identify anomalies that deviate from the majority of samples. Early anomaly detection algorithms use different metrics, such as distance-based [18] and density-

based [7], to measure the deviation degree among samples to detect anomalies. For example, isolation forest-based algorithms [25, 26] identify anomalous by building the binary search tree structure to isolate samples. In addition, anomaly detection algorithms based on one-class classification [40, 39, 37, 63] and autoencoder [2, 64] have also been in the spotlight. For the graph-structured data, most graph anomaly detection algorithms dedicate to detect anomalies (e.g., anomalous nodes [44, 13, 58] and edges [36, 57]) in a single graph [1].

The anomaly notions of graph-level anomalies are more diversified than those of node-level anomalies. For example, node-level anomaly detection methods usually regard nodes that betray network homophily[1] as anomalies [44]. However, at the graph level, nodes that can achieve an anomalous graph go far beyond the nodes that do not meet homophily. Node-level anomaly detection methods identify anomaly nodes from a micro perspective (e.g., neighborhood structures) while identifying anomalous graphs needs learning graphs at the macro level. Employing node-level anomaly detection methods to detect graph-level anomalies could capture some anomalies, but only a small part. Moreover, graph-level anomaly detection algorithms need to consider anomaly notions related to graph substructures. Hence, it is necessary to propose anomaly detection methods that are specialized in detecting graph-level anomalies.

**Graph-level Anomaly Detection.** To the best of our knowledge, there are only several preliminary research on graph-level anomaly detection. Zhao et al. [62] investigate a research question of "should we use the data for the graph classification task to evaluate graph-level anomaly detection models?", followed by which two graph-level anomaly detection algorithms [27, 35] combine Graph Neural Networks (GNNs) with knowledge distillation and one-class classification respectively to detect anomalous graphs.

Graph-level anomaly detection can be regarded as a special case of graph classification. There is a huge gap between the number of graphs belonging to the normal class versus the anomalous ones. But, graph classification algorithms usually assume that the number of graphs in each class is close. GKs [47, 16, 42, 6, 41] used to be the mainstream graph classification methods. Generally, GKs use the kernel function to calculate kernel values (i.e., similarity) between pair-wised graphs. Afterward, an off-the-shelf classifier is applied to the graph similarity matrix to perform the classification task. Graph classification algorithms based on GKs cannot learn graph representations explicitly and be optimized in an end-to-end fashion. In recent years, graph mining practitioners pay more attention to GNNs [8, 11, 4, 38]. GNNs with spatial graph convolution [20, 46, 19] are summarized as message-passing framework (MPNNs) [17, 54, 59, 56, 52]. In general, MPNNs update the representation of each node by iteratively aggregating information coming from their neighbors. To represent a graph, MPNNs apply the readout function or pooling operation on node embeddings. The spatial graph convolution that mix the representation of nodes and their neighborhoods to represent nodes will smooth node attributes [61, 4]. In addition, [31, 28, 60] proved that spatial graph convolution has limited power in distinguishing graph structures. Performing message passing between subgraphs can enable spatial graph convolution to capture more graph structural information. But such a solution exponentially consumes memory resources and computation complexities [3].

Graph classification methods mentioned above are likely to underfit anomalous graphs since they do not have any intrinsic mechanism to target the imbalanced distributions. In iGAD, a novel loss is presented for the imbalanced distributions by modeling PMI. In addition, different from existing graph-level anomaly methods [62, 27, 35] that employ the off-the-shelf MPNNs (e.g., GIN [54]) to represent graphs, iGAD can embed the information of anomalous attributes and substructures into graph representations.

**Imbalanced Learning.** Learning data with imbalanced distribution can be divided into data- and algorithm-level approaches. The former alleviates the imbalanced distribution of the original data by either under-sampling the majority [49] or over-sampling the minority [9]. However, modifying the original distribution of data will bring a lot of drawbacks inevitably, such as over-fitting and discarding important samples. There are different types of approaches to address the issue at the algorithm level. The typical approaches include employing post-hoc correction [30], aligning data distribution, [53], and modifying the loss function [24, 22]. In the real-world application, considering the different consequences of false-positive identification (e.g., predicting an inactive chemical compound to be

---

[1]In a graph with high homophily, nodes tend to have the same label and similar features as their neighbors. Node-level anomaly detection methods focus on the neighborhood structure and usually identify the nodes which do not have the same label or similar features with most of their neighbors as anomalies.

active) and false-negative identification (e.g., predicting a cancer patient as a healthy individual), the situation will be more complicated [33]. In this paper, iGAD introduces PMI to the cross-entropy loss function to solve this problem.

## 3 Preliminaries

**Notations.** A graph dataset consisting of $N$ graphs can be denoted as $\mathcal{G} = \{G_1, ..., G_N\}$, where $G_i = \{V_i, E_i\}$ is an unweighted and undirected graph in $\mathcal{G}$, $V_i$ and $E_i$ are the node and the edge set, respectively. The topology of $G_i$ can be modeled as an adjacency matrix $\mathbf{A}_i \in \{0, 1\}^{n \times n}$, where $\mathbf{A}_i(u, v) = 1$ if there is an edge between nodes $u$ and $v$, otherwise 0. $\mathbf{A}_i$ does not consider the self-loop, $\mathbf{A}_i(u, u) = 0, \forall u \in V_i$. We use $\mathbf{x}_u \in \mathbb{R}^c$ to denote the attribute vector of node $u \in V_i$. $\mathbf{X}_i \in \mathbb{R}^{n \times c}$ is the attribute matrix of $G_i$. $G_i' = \{V_i', E_i'\}$ can be regarded as a substructure in $G_i$, iff $V_i' \subseteq V_i$ and $E_i' \subseteq E_i$. $\mathcal{Y}$ is the label set of $\mathcal{G}$. $G_i$ is labeled by $y_i \in \{0, 1\}$, where 1 represents *anomalous* and 0 represents *normal*.

**Problem Definition.** This paper concentrates on the supervised graph-level anomaly detection problem. Given the training set $\mathcal{T} = \{(G_1, y_1), (G_2, y_2), ...\}$, we aim to train a model to predict the normal/anomalous proprieties of unseen graphs. $\mathcal{T}$ contains $N_1$ *anomalous* graphs and $N_0$ *normal* graphs ($N_1 \ll N_0$). The imbalanced ratio of $\mathcal{T}$ is defined as $\delta$ ($\delta = \frac{N_1}{N_0 + N_1}$).

**Spatial Graph Convolution.** A graph convolution layer updates node representations by iteratively aggregating information coming from its one-hop neighborhood. The formulation is:

$$\mathbf{h}^{(k+1)}(u) = \sigma \left( f_{\text{COMBINE}}(\mathbf{h}^{(k)}(u), f_{\text{AGGREGATE}}(\{\mathbf{h}^{(k)}(v), v \in \mathcal{N}(u)\})) \right), \mathbf{h}^{(0)}(u) = \mathbf{x}_u, \quad (1)$$

where $0 \leq k \leq K - 1$, $\mathcal{N}(u)$ denotes the 1-hop neighborhood of node $u$. $f_{\text{AGGREGATE}}$ denotes how to aggregate neighbors of node $u$, and $f_{\text{COMBINE}}$ defines how to combine the immediate representations of node $u$ and its neighbors. $\sigma$ is the activation function. Generally, the final representation of the node $u$ is $\mathbf{h}^{(K)}(u)$.

**Random Walk Kernels (RWK).** RWK measures graph similarities by counting common random walk sequences among graphs. There are two graphs $G_1 = \{V_1, E_1\}$ and $G_2 = \{V_2, E_2\}$. $G_\times = \{V_\times, E_\times\}$ denotes the direct product graph about $G_1$ and $G_2$, where $V_\times = \{(u, u') | u \in V_1, u' \in V_2\}$ and $E_\times = \{((u, u'), (v, v')) | (u, v) \in E_1, (u', v') \in E_2\}$. Performing random walking on $G_\times$ is equivalent to doing random walking on $G_1$ and $G_2$ simultaneously. The adjacency matrix of direct product graph $\mathbf{A}_\times$ can be obtained by the Kronecker product operation $\otimes$ on $\mathbf{A}_1$ and $\mathbf{A}_2$, that is $\mathbf{A}_\times = \mathbf{A}_1 \otimes \mathbf{A}_2$. The number of $l$-length common walk sequences between $G_1$ and $G_2$ is $\kappa^{(l)}(G_1, G_2) = \mathbf{q}_\times^\mathsf{T}(\mathbf{A}_\times)^l \mathbf{p}_\times$, where $\mathbf{p}_\times$ and $\mathbf{q}_\times$ are the vector about the starting and stopping probabilities of random walks on $G_\times$, respectively. The elements in $\mathbf{p}_\times$ and $\mathbf{q}_\times$ are all ones when there is no prior knowledge. Overall, the RWK about $G_1$ and $G_2$ can be defined as:

$$\mathcal{K}(G_1, G_2) := \sum_{l=0}^{\infty} \kappa^{(l)}(G_1, G_2). \quad (2)$$

## 4 iGAD

In this section, we will describe iGAD in detail (see Figure 1 for an illustration). iGAD contains three key designs: (1) anomalous graph attribute-aware graph convolution; (2) anomalous substructure-aware deep RWK; and (3) PMI-based loss function. The first two designs encode the anomalous information on graph attributes and substructures into graph embeddings. The loss function makes the training on iGAD not completely dominated by normal graphs.

### 4.1 iGAD: Anomalous Attribute-aware Graph Convolution

As aforementioned in Section 1, the attributes of particular nodes can achieve an anomalous graph. The graph convolution which iteratively updates node representations by performing feature transformation on the concatenated information about nodes and their neighborhoods will make node representations over-smoothed [61]. Under such a graph convolution mechanism, only limited anomalous information can be manifest in corresponding node representations. We argue that making each

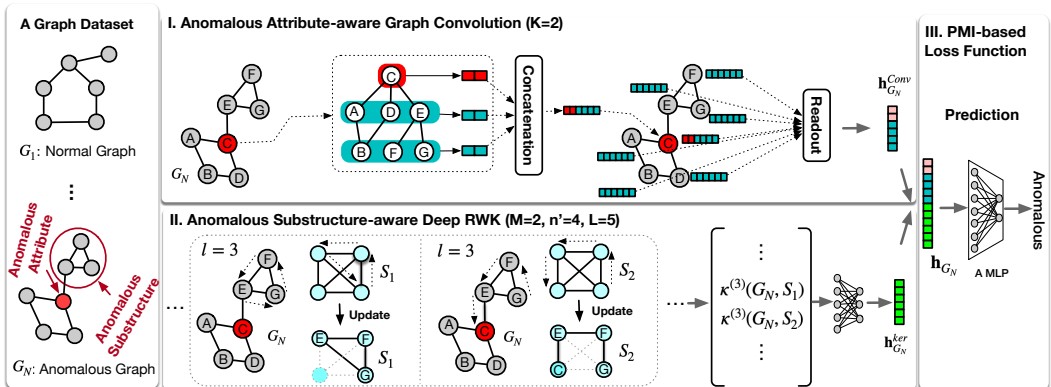

Figure 1: Taking learning the graph $G_N$ in a graph dataset as an example, we illustrate the framework of iGAD with three designs. I. Anomalous Attribute-aware Graph Convolution, which concatenates the independent encoding on nodes and their k-hop neighborhoods to represent nodes in $G_N$. To obtain the graph representation $\mathbf{h}_{G_N}^{\mathrm{CONV}}$, a readout function is applied to all node representations. II. The Anomalous Substructure-aware Deep RWK, which performs random walks on two learnable anomalous substructures (i.e., $S_1$ and $S_2$) and the input graph $G_N$ simultaneously. Deep RWK obtains $\mathbf{h}_{G_N}^{\mathrm{KER}}$ based on kernel values $\kappa^{(l)}(G_N, S_m), l = \{1, .., L\}, m = \{1, ..., M\}$. III. The PMI-based Loss Function. iGAD concatenates $\mathbf{h}_{G_N}^{\mathrm{CONV}}$ and $\mathbf{h}_{G_N}^{\mathrm{KER}}$ to obtain the final graph representation $\mathbf{h}_{G_N}$. Finally, a MLP is applied to $\mathbf{h}_{G_N}$ to predict $G_N$. The model training is guided by a PMI-based loss function.

node's attributes evolve independently over the graph convolution process is critical for capturing anomalous attributes. To achieve it, anomalous attribute-aware graph convolution concatenates the independent encoding on the node itself and the independent encoding on its $k$-hop neighborhood $(1 \leq k \leq K)$ to represent each node in a graph. Formally,

$$\mathbf{h}_u = f_{\mathrm{COMBINE}}\left(\mathbf{h}^{(0)}(u), \mathbf{h}^{(1)}(u), ..., \mathbf{h}^{(K)}(u)\right), \tag{3}$$

where $\mathbf{h}_u$ is the final node representation of the node $u$, $\mathbf{h}^{(k)}(u)$ is the representation of node $u$'s $k$-hop neighborhood, and $f_{\mathrm{COMBINE}}$ is the concatenation operation. $\mathbf{h}^{(0)}(u)$ and $\mathbf{h}^{(k)}(u)$ are formulated as:

$$\mathbf{h}^{(0)}(u) = \mathrm{MLP}(\mathbf{x}_u), \tag{4}$$

and

$$\mathbf{h}^{(k)}(u) = \sigma(\mathbf{A}_i \mathbf{H}_i^{(k-1)} \mathbf{W}^{(k)})_u, \tag{5}$$

respectively, where $\mathbf{H}_i^{(0)} = \mathbf{X}_i$, MLP denotes a multi-layer perceptron, $\sigma$ represents the activation function, and $\mathbf{W}^{(k)}$ is the trainable weight matrix. To obtain the entire graph representation $\mathbf{h}_{G_i}^{\mathrm{CONV}}$, we apply the permutation invariant Readout function $\mathcal{R}(\cdot)$ on all nodes in the graph $G_i$. That is,

$$\mathbf{h}_{G_i}^{\mathrm{CONV}} = \mathcal{R}\left(\{\mathbf{h}_u | u \in V_i\}\right), \tag{6}$$

where $\mathcal{R}(\cdot)$ can be the sum-based or mean-based global pooling operation.

## 4.2  iGAD: Anomalous Substructure-aware Deep RWK

Generally, the neighborhood aggregation mechanism adopted by existing GNNs only depends on local neighborhood structures to perform message passing [60]), which leads to the deficiency of graph convolution in distinguishing substructures [3, 10]. To make iGAD to be aware of anomalous substructures, we resort to RWK that can explore diverse substructures through random walking sequences. There is a pioneering work [32] that represents graphs by calculating RWK values between input graphs and some substructures. It inspires us to embed information about anomalous substructures into graph representations by performing random walks in input graphs and learnable anomalous substructures simultaneously.

We present an anomalous substructure-aware deep RWK, which models elements in the adjacency matrices of anomalous substructures as learnable parameters. Supposed that $\{S_1, ..., S_M\}$ is the set

of $M$ anomalous substructures, and the size of $S_m$ is $n'$. $\mathbf{P}_m \in \mathbb{R}^{n' \times n'}$ is the adjacency matrix of $S_m$, where the matric element $\mathbf{P}_m(u', v')$ denotes the possibility that an edge exists between nodes $u'$ and $v'$ in $S_m$. As the adjacency matrix is symmetric, $\mathbf{P}_m$ contains $\frac{n'(n'-1)}{2}$ parameters. Referring to Eq. 2 and supposing that the starting and the stopping possibilities on each node are all ones, the number of $l$-length common random walk sequences between the input graph $G_i$ and the anomalous substructure $S_m$ is:

$$\kappa^{(l)}(G_i, S_m) = \mathbf{1}^\mathsf{T}(\mathbf{A}_i \otimes \mathbf{P}_m)^l \mathbf{1} = \sum_i \sum_j \left[(\mathbf{P}_m)^l \mathbf{E}(\mathbf{A}_i)^l\right]_{i,j}, \tag{7}$$

where $\mathbf{1} \in \mathbb{R}^{n'n}$ is a column vector full of 1, and $\mathbf{E} \in \mathbb{R}^{n' \times n}$ is a matrix in which all elements are 1. The proof of Eq. 7 can be found in Appendix A.1. RWK with infinite random walk lengths will often meet the halting issue [43]. Therefore, iGAD only considers the random walk length from 1 to $L$. As the size of anomalous substructures ($n'$) is small, optimizing $(\mathbf{P}_m)^l$ is not necessary. The adjacency matrix $\mathbf{A}_i$ is sparse, and we compute $(\mathbf{A}_i)^l$ by the sparse matrix multiplication in PyTorch [34]. That is $(\mathbf{A}_i^s)^l = \mathbf{A}_i^s(\mathbf{A}_i^s...(\mathbf{A}_i^s(\mathbf{A}_i))...)$, where $\mathbf{A}_i^s$ is the sparse version of $\mathbf{A}_i$. Afterward, deep RWK values $\kappa^{(l)}(G_i, S_m)$ about $G_i$ and $S_m$, $m = \{1, .., M\}, l = \{1, .., L\}$ will be assembled into a vector $\mathbf{a}_{G_i}$. That is,

$$\mathbf{a}_{G_i} = \left(\kappa^{(1)}(G_i, S_1), ..., \kappa^{(1)}(G_i, S_M); ...; \kappa^{(L)}(G_i, S_1), ..., \kappa^{(L)}(G_i, S_M)\right), \tag{8}$$

where $\mathbf{a}_{G_i} \in \mathbb{R}^{ML}$. The graph representation $\mathbf{h}_{G_i}^{\text{KER}}$ based on deep RWK is:

$$\mathbf{h}_{G_i}^{\text{KER}} = \text{MLP}(\mathbf{a}_{G_i}). \tag{9}$$

The final graph representation $\mathbf{h}_{G_i}$ for $G_i \in \mathcal{G}$ can be obtained by concatenating $\mathbf{h}_{G_i}^{\text{CONV}}$ and $\mathbf{h}_{G_i}^{\text{KER}}$ together. The anomalous attribute-aware graph convolution and anomalous substructure-aware deep RWK are welded into a graph neural network together through $\mathbf{h}_{G_i}$.

## 4.3 iGAD: PMI-based Loss Function

The number of normal graphs is much more than that of anomalous graphs. Under the imbalance nature of the anomaly problem, normal graphs will dominate model training. Targeting this issue, we propose a new loss function that can model the PMI between input graphs and their anomalous/normal properties. Specifically, we use $f(\mathcal{G}; \Theta)$ to denote the proposed model iGAD, where $\Theta$ represents model parameters. Modeling the PMI between the input graph $G_i$ and its anomalous/normal properties can be formulated as:

$$f_{y_i}(G_i; \Theta) \sim \log \frac{P_\Theta(G_i, y_i)}{P(G_i)P(y_i)}, \tag{10}$$

where the right part represents the PMI between $G_i$ and $y_i$, $P(y_i)$ is the proportion of anomalous graphs or normal graphs in the training set $\mathcal{T}$. Compared with most existing loss functions, especially the loss function that models the conditional distribution about $y_i$ given $G_i$ (i.e., $f_{y_i}(G_i; \Theta) \sim P_\Theta(y_i|G_i)$), PMI enables iGAD to give priority to learning the information that reveals the essential correlation between graphs and their labels. Eq. 10 can be reformatted as:

$$\log P_\Theta(y_i|G_i) \sim f_{y_i}(G_i, \Theta) + \log P(y_i). \tag{11}$$

After softmax normalization, the distribution of $y_i$ given $G_i$ is formulated as:

$$P_\Theta(y_i|G_i) = \frac{e^{f_{y_i}(G_i;\Theta)+\log P(y_i)}}{\sum_{y_i} e^{f_{y_i}(G_i;\Theta)+\log P(y_i)}}. \tag{12}$$

Incorporating $P_\Theta(y_i|G_i)$ in to the framework of the cross-entropy loss function, we can obtain the PMI-based loss function $\mathcal{L}$:

$$\mathcal{L} = \frac{1}{N} \sum_{i=1}^{N} -\log P_\Theta(y_i|G_i). \tag{13}$$

Specifically, iGAD predict the graph $G_i$:

Table 1: Statistical information of four real-world datasets. We regard the one-hot encoding of node labels (e.g., carbon (C), nitrogen (N), oxygen (O)) as node attributes.

| DATASET | #GRAPHS ($N$) | #ANOMALOUS GRAPHS | $\delta$ | AVG. #NODES (AVG. $n$) | AVG. #EDGES (AVG. $|E|$) | # NODE LABELS ($c$) | TUMOR |
|---------|---------------|-------------------|----------|------------------------|--------------------------|---------------------|-------|
| SW-620 | 40,532 | 2,411 | 5.95% | 26.05 | 28.08 | 65 | COLON |
| MOLT-4 | 39,765 | 2,904 | 7.90% | 26.07 | 28.13 | 64 | LEUKEMIA |
| MCF-7 | 27,770 | 2,293 | 8.26% | 26.39 | 28.52 | 46 | BREAST |
| PC-3 | 27,509 | 1,568 | 5.70% | 26.35 | 28.49 | 45 | PROSTATE |

$$y_i^* = \begin{cases} \underset{y_i}{\text{argmax}}\, f_{y_i}(G_i; \Theta) + \log P(y_i), & \text{during training.} \\ \underset{y_i}{\text{argmax}}\, f_{y_i}^*(G_i; \Theta), & \text{during testing.} \end{cases} \tag{14}$$

The reason that we use $\text{argmax}_{y_i} f_{y_i}^*(G_i; \Theta)$ to predict a graph in test time can be found in Appendix A.2.

Many other loss functions target the difficulties brought by imbalanced distributions, such as the cost-sensitive loss function that assigns or learns different weights to the majority and the minority samples [29, 50, 12]. Compared with them, the PMI-based loss function does not introduce any new parameters that need to be carefully tuned. It comes in the form of the most widely used cross-entropy loss function and gives prediction results that can achieve maximized PMI between inputs and predictions.

## 4.4 Algorithm Description & Computational Complexity

The pseudocode of iGAD is included in Algorithm 1 in Appendix A.1. We divide graphs in $\mathcal{G}$ into different mini-batches to train iGAD [23]. But for illustrating how iGAD predicts a graph more clearly, the batch size is set as 1 in Algorithm 1. Given the training set, we initialize anomalous substructures and other parameters (Line 1 in Algorithm 1) and calculate the proportion of normal and anomalous graphs (Line 2). For each graph, iGAD learns its graph representation (Lines 3 to 15). Based on the graph representation, a MLP equipped with the PMI-based loss function gives predictions to graphs (Lines 16 and 17). At the back-propagation stage, anomalous substructures and other parameters will get updated. At each epoch, deep RWK performs random walking on input graphs and anomalous substructures that get updated during the previous epoch.

**Lemma 1** *The computational complexity of iGAD is $\mathcal{O}(n + |E|)$, where $n$ and $|E|$ are the number of nodes and edges, respectively. (The detailed analysis can be found in Appendix A.3).*

# 5 Experiments

In this section, we conduct a series of experiments to study the performance of iGAD on the graph-level anomaly detection task. Specifically, we investigate the following six questions:

Q1. How about the performance of iGAD on the graph-level anomaly detection task?
Q2. Does iGAD benefit from the PMI-based loss function?
Q3. How about the performance of the anomalous graph attribute-aware graph convolution in iGAD?
Q4. How about the performance of iGAD when it utilizes pre-defined anomalous substructures?
Q5. Can the anomalous substructure-aware deep RWK in iGAD learn anomalous substructures?
Q6. How does the maximum length of random walk sequences affect the performance of iGAD?

The questions from Q2 to Q4 can be regarded as the ablation study on our three main design elements, they are: (a). cross-entropy loss function versus the proposed PMI-based loss function; (b). general graph convolution versus the proposed graph convolution; (c). RWK versus the proposed deep RWK.

Table 2: Graph-level anomaly detection performance (%) of five graph classification methods, their GAD variants, iGAD, and iGAD's two variants on four datasets. The value behind $\pm$ denotes the standard deviation (mean $\pm$ std) over different data splits. The best result is highlighted in gray. The description about evaluation metrics *AUC*, *Recall*, *Recall(A)*, and *F-score* is in Appendix B.1.

| DATA | METRICS | GCN | GCN-GAD | GIN | GIN-GAD | DGCNN | DGCNN-GAD | SOPOOL | SOPOOL-GAD | RWGNN | RWGNN-GAD | iGAD-1 | iGAD-2 | iGAD |
|------|---------|-----|---------|-----|---------|-------|-----------|--------|------------|-------|-----------|--------|--------|------|
| SW-620 | AUC | 74.90±0.74 | 76.48±1.51 | 78.61±2.85 | 76.42±2.80 | 80.06±0.42 | 79.36±1.28 | 75.51±5.06 | 76.89±2.07 | 73.37±0.36 | 74.92±1.97 | 85.40±0.93 | 85.77±0.72 | 85.82±0.69 |
| | RECALL | 51.79±0.47 | 69.65±1.39 | 56.62±3.97 | 70.38±2.18 | 56.02±1.16 | 72.65±0.63 | 56.01±2.29 | 69.97±1.69 | 51.43±0.87 | 68.32±1.76 | 78.60±0.97 | 79.08±1.31 | 79.64±0.83 |
| | RECALL(A) | 3.78±0.95 | 67.26±1.88 | 14.40±8.57 | 64.11±6.83 | 12.74±2.41 | 67.84±1.78 | 13.28±5.05 | 62.32±6.23 | 3.11±1.82 | 63.82±5.40 | 74.23±3.66 | 73.24±2.52 | 74.27±2.99 |
| | F-SCORE | 52.00±0.87 | 52.44±1.32 | 58.47±5.43 | 55.00±2.82 | 58.73±1.64 | 56.11±0.89 | 58.11±2.86 | 55.63±4.78 | 51.31±1.62 | 52.47±3.17 | 61.82±2.22 | 63.36±1.58 | 63.68±1.56 |
| MOLT-4 | AUC | 72.55±0.52 | 72.88±0.99 | 75.86±1.60 | 75.43±2.82 | 76.50±0.60 | 77.43±0.73 | 75.11±0.97 | 74.50±1.69 | 71.30±1.23 | 71.51±0.70 | 81.59±1.10 | 82.73±1.12 | 83.59±1.07 |
| | RECALL | 51.26±0.54 | 66.84±0.66 | 56.87±6.98 | 69.40±2.47 | 55.35±1.27 | 70.14±0.69 | 55.31±3.74 | 69.12±1.50 | 51.39±1.41 | 65.84±1.27 | 74.97±1.61 | 76.33±1.18 | 76.82±0.82 |
| | RECALL(A) | 2.74±1.12 | 67.32±3.04 | 17.71±9.02 | 64.36±7.14 | 11.69±2.87 | 65.76±1.97 | 12.23±9.34 | 63.44±6.55 | 3.18±3.21 | 61.24±3.89 | 68.18±2.72 | 70.54±4.28 | 72.10±2.47 |
| | F-SCORE | 50.53±1.03 | 51.25±1.50 | 55.43±6.52 | 55.86±2.00 | 57.33±1.76 | 56.22±1.32 | 56.20±3.64 | 56.02±3.17 | 50.67±2.64 | 52.69±2.60 | 62.22±0.48 | 63.12±1.31 | 63.30±1.17 |
| PC-3 | AUC | 75.36±2.13 | 75.62±1.22 | 78.44±1.67 | 76.95±1.74 | 79.15±1.84 | 78.90±1.70 | 69.37±1.53 | 78.24±1.97 | 76.27±0.86 | 76.10±1.92 | 85.44±1.20 | 85.83±1.02 | 86.04±1.14 |
| | RECALL | 50.72±0.59 | 69.46±1.57 | 57.43±4.61 | 71.49±1.27 | 52.88±0.28 | 72.53±1.40 | 56.78±3.79 | 71.44±1.53 | 51.02±1.53 | 69.90±1.74 | 78.31±0.76 | 79.94±0.64 | 79.59±0.41 |
| | RECALL(A) | 1.53±1.26 | 65.29±5.69 | 16.31±10.47 | 68.22±6.76 | 6.05±0.57 | 70.06±5.19 | 15.67±8.98 | 69.60±9.54 | 2.17±3.26 | 66.62±3.03 | 70.45±3.29 | 75.35±0.74 | 75.69±1.64 |
| | F-SCORE | 49.99±1.63 | 52.76±1.11 | 59.07±4.15 | 54.09±3.04 | 54.01±0.48 | 54.46±1.70 | 57.80±3.74 | 51.76±5.27 | 50.44±2.76 | 52.72±1.54 | 61.17±2.43 | 62.97±1.40 | 63.50±0.73 |
| MCF-7 | AUC | 72.70±1.05 | 73.89±0.99 | 69.54±1.15 | 75.90±1.44 | 76.41±0.81 | 76.83±0.48 | 75.64±2.17 | 76.69±1.33 | 70.47±1.26 | 73.25±1.56 | 83.02±0.46 | 83.49±1.03 | 83.22±0.64 |
| | RECALL | 50.42±0.28 | 67.14±1.13 | 57.08±3.82 | 69.40±2.32 | 55.11±1.11 | 70.74±0.63 | 56.82±3.57 | 68.88±1.07 | 52.09±1.06 | 66.99±1.91 | 75.80±0.81 | 76.81±0.97 | 76.77±0.47 |
| | RECALL(A) | 0.96±0.63 | 61.79±2.14 | 16.69±9.92 | 59.48±10.14 | 11.07±2.33 | 69.37±3.05 | 15.34±8.20 | 63.92±7.30 | 4.97±5.24 | 57.73±11.16 | 67.67±2.54 | 71.59±2.19 | 73.38±3.59 |
| | F-SCORE | 48.76±0.59 | 53.37±1.89 | 58.27±3.20 | 58.71±2.53 | 56.99±1.70 | 55.74±1.54 | 56.82±3.57 | 54.82±3.00 | 51.65±3.46 | 56.06±2.93 | 61.56±1.89 | 63.75±1.18 | 64.70±2.58 |

## 5.1 Experimental Setup

**Datasets.** SW-610, MOLT-4, PC-3, and MCF-7 are four real-world graph datasets. These datasets are collected from PubChem[2], which records a tremendous amount of chemical compounds and their anti-cancer activity testing results ("active" or "inactive") on different types of cancer cell lines. We regard the chemical compounds with the outcome "active" as anomalous, and "inactive" as normal. For example, the graph dataset SW-620 records 40,532 chemical formulas and their anti-cancer activity testing results on the colon cancer cell line. Chemical compounds that exhibit antibody activity against colon cancer are labeled as anomalous graphs, otherwise normal graphs. The statistical information about these four datasets is shown in Table 1.

**Baselines & Experimental Settings.** To investigate Q1, we compare iGAD with the following baseline models: (1) GCN[20] + Readout function (abbreviated GCN) and GIN [54]. These two algorithms employ spatial graph convolution to learn node representation and apply the readout function to node representations to obtain graph representations and classify graphs; (2) DGCNN [59] and SOPool [52]. In these two models, spatial graph convolution and the graph pooling operation are employed to downsize graphs and complete the graph classification task; (3) RWGNN [32]. It employs random walk sequences to represent graphs and then classify graphs based on graph representations.

Table 3: The performance (%) of two graph-level anomaly detection methods (i.e., GLo-calKD and OCGTL) and iGAD *w.r.t.* AUC.

| DATA | GLocalKD | OCGTL | iGAD |
|------|----------|-------|------|
| SW-620 | 64.14±0.92 | 67.69±0.02 | 85.82±0.69 |
| MOLT-4 | 61.43±1.26 | 57.42±2.38 | 83.59±1.07 |
| PC-3 | 64.79±1.22 | 68.42±1.73 | 86.04±1.14 |
| MCF-7 | 61.43±1.26 | 64.92±1.92 | 83.22±0.64 |

All the above baselines are supervised and use the cross-entropy loss function; (4) The graph-level anomaly detection methods GLocalKD [27] and OCGTL [35]. The former adopts random distillation to detect the locally- and globally-anomalous graphs and the latter combines the one-class classification and graph transformation learning to identify anomalous graphs. Both of them only use normal graphs to train models. They learn anomaly scores for graphs to measure each graph' anomaly degree and do not provide any criterion to determine whether a graph is anomalous or not.

For the fairness of experiments and to answer Q2, we also compare iGAD with GCN-GAD, GIN-GAD, DGCNN-GAD, SOPool-GAD, and RWGNN-GAD. These algorithms are the graph anomaly detection (GAD) variants of GCN, GIN, DGCNN, SOPool, and RWGNN. We replace the cross-entropy loss in these algorithms with the proposed PMI-based loss function. To answer Q3, we consider a variant of iGAD —iGAD-1 that only utilizes the anomalous graph attribute-aware graph convolution to represent graphs and detect anomalous graphs. To answer Q4, we consider the other variant named iGAD-2 in which the random walking performs on input graphs and pre-defined anomalous graph substructures simultaneously. Specifically, we use 3-star, triangle, tailed triangle,

---

[2]https://pubchem.ncbi.nlm.nih.gov

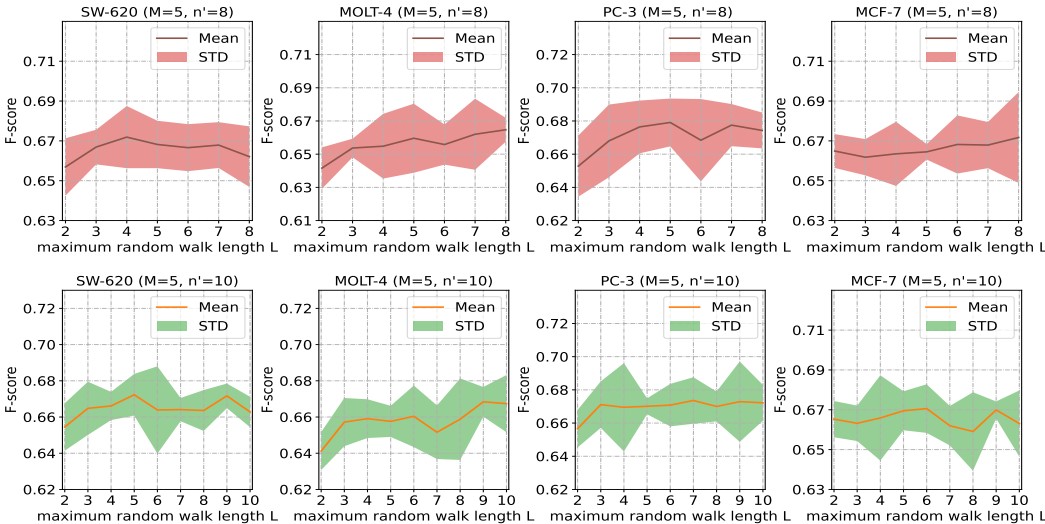

Figure 3: The impact of different maximum random walk length $L$ on the performance of iGAD. The first row reports the performance of iGAD under the setting $M$=5 and $n'$=8. The second row reports the performance of iGAD under the setting $M$=5 and $n'$=10.

and 4-cycle graphlets as pre-defined graph anomalous substructures (see Figure 4 in Appendix B for these substructures). Information about parameter setting and algorithm implementation can be found in Appendix B.1. We report experimental results under 5-fold cross-validation.

## 5.2 Experimental Results

This section answers Q1 - Q6 by A1 - A6, respectively. We put extra experiments in Appendix B.2.

**A1.** iGAD demonstrates superior performance in identifying graph-level anomalies. Observed from Table 2: 1) Compared with the graph classification baselines GCN, GIN, DGCNN, SOPool, and RWGNN, iGAD demonstrates significant advantages on four real world datasets in terms of all evaluation metrics; 2) iGAD also beats the above baselines' GAD variants, GCN-GAD, GIN-GAD, DGCNN-GAD, SOPool-GAD, and RWGNN-GAD in terms of all evaluation metrics. In addition, as shown in Table 3, iGAD outperforms two graph-level anomaly detection methods GLocalKD and OCGTL concerning AUC.

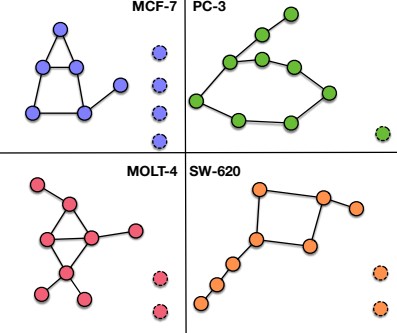

Figure 2: The anomalous substructures learned by iGAD under the experimental setting $M$=1, $n'$=10, $L$=5. We only draw the edge $(u', v')$ with the learned weight $\mathbf{P}_m(u', v')$ over 0.5.

**A2.** PMI-based loss function can capture essential correlations between graphs and their anomalous/normal proprieties even if the data distribution is imbalanced. As shown in Table 2, compared with GNNs with the cross-entropy loss function (GCN, GIN, DGCNN, SOPOOL, and RWGNN), their GAD variants equipped with the newly proposed PMI-based loss function (GCN-GAD, GIN-GAD, DGCNN-GAD, SOPOOL-GAD, and RWGNN-GAD) witness a significant performance improvement regarding *Recall* and *Recall(A)* over all datasets. For example, the value of *Recall* and *Recall(A)* achieved by DGCNN over the dataset SW-620 are $56.02_{\pm 1.16}$ and $12.74_{\pm 2.41}$ respectively. In contrast, $72.65_{\pm 0.63}$ and $67.84_{\pm 1.78}$ are reported by DGCNN-GAD. PMI-based loss function can greatly improve GNN-based graph classification model in datasets with highly imbalanced class distributions.

**A3.** The anomalous attribute-aware graph convolution can embed anomalous attributes into graph representations and improve graph-level anomaly detection. As shown in Table 2, iGAD-1 significantly outperforms all baseline algorithms on four graph datasets in terms of *AUC*, *Recall*, and *F-score*.

Concerning *Recall(A)*, on the dataset MCF-7, iGAD-1 has a little performance degradation compared with DGCNN-GAD. Specifically, DGCNN-GAD achieves $69.37_{\pm3.05}$ on MCF-7, while iGAD-1 gets $67.67_{\pm2.54}$. But iGAD-1 beats DGCNN-GAD on other datasets in terms of *Recall(A)*.

**A4.** iGAD is also effective when it is equipped with pre-defined anomalous substructures. As shown in Table 2, iGAD-2 demonstrates significant advantages against all baseline models. Moreover, iGAD-2 also beats iGAD-1 concerning all metrics over datasets MOLT-4, PC-3, and MCF-7. iGAD-2 does not outperform than iGAD on all datasets concerning *AUC*, *Recall(A)*, and *F-score*. But, on PC-3 and MCF-7, iGAD-2 has a slight lead over iGAD regarding *Recall*. The experiment results demonstrate that, in most cases, the performance of iGAD equipped with a deep RWK is better than iGAD-2 using pre-defined anomalous substructures and algorithms that do not consider anomalous substructures. Based on the above analysis, we can conclude that comparing the random walk sequences between the input graph and pre-defined substructure can improve graph-level anomaly detection to a certain extent, but the algorithm that can automatically learn the anomalous substructures is better.

**A5.** The anomalous substructures learned by iGAD are demonstrated in Figure 2. We can observe that anomalous graph substructures on four datasets are composed of the triangle, the tailed triangle, and the 4-cycle graphlets. It explains why iGAD-2 can achieve competitive results compared with iGAD (as shown in Table 2 and **A4**). We can draw a preliminary conclusion that the graph substructures having the triangle, the tailed triangle, and the 4-cycle graphlets are important for distinguishing anomalous graphs from normal graphs on these four graph datasets.

**A6.** The performance of iGAD under different maximum random walk lengths $L$ is shown in Figure 3. Under the setting that iGAD with 5 anomalous substructures and each of substructures has 8 nodes, on these four graph datasets SW-620, MOLT-4, PC-3, and MCF-7, iGAD can achieve the highest *F-score* value when the maximum random walk length ranges from 4 to 8. In addition, when each anomalous substructure contains 10 nodes, iGAD can reach the best *F-score* score at the maximum random walk length from 5 to 9. The results verify that appropriately increasing the maximum length of random walking can improve the performance of iGAD since more much longer random walk sequences can capture more graph structure information.

Finally, we discuss the limitations of the proposed graph-level anomaly detection model iGAD. In practice, the nodes and/or substructures at a particular position in a graph can also be the reason for the graph's anomalous properties. For example, an Alcohol compound can be obtained by replacing the hydrogen atom at the end of the Alkane compound with a hydroxy group (i.e., -OH). But, iGAD cannot capture the position information of nodes and substructures in graphs. In addition, iGAD is a supervised algorithm. It may not be able to achieve superior performance in identifying the graph-level anomalies that follow the anomaly distributions that have never been exposed in the training graph anomalies. In the future, we will propose specific solutions to this problem.

# 6  Conclusion

This paper proposes a method named iGAD for graph-level anomaly detection. We highlight the challenges brought by imbalanced distributions and various anomaly notions in graph attributes and substructures. In addition, the significant differences between detecting anomalous graphs in a graph dataset and identifying anomalous nodes in a single graph are also introduced in detail. iGAD treats graph-level anomaly detection as a special case of graph classification and learns binary labels for graphs. In iGAD, the anomalous attribute-aware graph convolution and anomalous substructure-aware deep RWK embed graph-level anomaly information into graph representations. A PMI-based loss function guides iGAD to identify anomalous samples from a large number of normal samples. We do extensive experiments on four real-world graph datasets, and iGAD achieves superior performance on the graph-level anomaly detection task.

# Acknowledge

We greatly appreciate the constructive and insightful comments from reviewers. This work is funded by the Australian Research Council Discovery Early Career Researcher Award (ARC DECRA) Project (No. DE200100964) and the NSFC (No. 61872360).

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
