# Appendix

This document is the appendix of paper "Dual-discriminative Graph Neural Network for Imbalanced Graph-level Anomaly Detection".

# A  Algorithm

This section introduces the deduction about Eq. (7) (see **A.1**) and the explanation about Eq. (14) (see **A.2**). In addition, the time complexity analysis of iGAD is shown in **A.3**.

## A.1

**Proposition 1.** Let the matrix $\mathbf{A} \in \mathbb{R}^{n \times m}$, $\mathbf{B} \in \mathbb{R}^{m \times s}$, and $\mathbf{C} \in \mathbb{R}^{s \times o}$, and $vec(\cdot) : \mathbb{R}^{m \times n} \to \mathbb{R}^{mn}$ denotes the vectorization operation on a matrix. Then, we can get:

$$vec(\mathbf{ABC}) = (\mathbf{C}^{\mathsf{T}} \otimes \mathbf{A})vec(\mathbf{B}), \tag{15}$$

where $\otimes$ denotes the Kronecker product. (The proof can be found in 7.1.9 in [1].)

**Proposition 2.** Let the matrix $\mathbf{A} \in \mathbb{R}^{n \times m}$, $\mathbf{B} \in \mathbb{R}^{o \times s}$, $\mathbf{C} \in \mathbb{R}^{m \times d}$, $\mathbf{D} \in \mathbb{R}^{s \times b}$. Then, we can get:

$$(\mathbf{A} \otimes \mathbf{B})(\mathbf{C} \otimes \mathbf{D}) = (\mathbf{AC}) \otimes (\mathbf{BD}), \tag{16}$$

where $\otimes$ denotes the Kronecker product. (The proof can be found in 7.1.6 in [1].)

**Proposition 3.** Let the matrix $\mathbf{A} \in \mathbb{R}^{n \times n}$, $\mathbf{B} \in \mathbb{R}^{m \times m}$, and $l \geq 1$. Then, we can get:

$$(\mathbf{A} \otimes \mathbf{B})^l = (\mathbf{A})^l \otimes (\mathbf{B})^l, \tag{17}$$

where $\otimes$ denotes the Kronecker product, $(\cdot)^l$ denotes matrix exponentiation.

**Proof of Proposition 3.** Based on **Proposition 2**, we can get $(\mathbf{A} \otimes \mathbf{B})(\mathbf{A} \otimes \mathbf{B}) = \mathbf{A}^2 \otimes \mathbf{B}^2$. Then:

$$\begin{aligned}
(\mathbf{A} \otimes \mathbf{B})^l &= (\mathbf{A} \otimes \mathbf{B})(\mathbf{A} \otimes \mathbf{B})(\mathbf{A} \otimes \mathbf{B})^{l-2} \\
&= (\mathbf{A}^2 \otimes \mathbf{B}^2)(\mathbf{A} \otimes \mathbf{B})^{l-2} \\
&= (\mathbf{A}^2 \otimes \mathbf{B}^2)\underbrace{(\mathbf{A} \otimes \mathbf{B})...(\mathbf{A} \otimes \mathbf{B})}_{l-2} \\
&= (\mathbf{A}^2 \otimes \mathbf{B}^2)(\mathbf{A} \otimes \mathbf{B})\underbrace{(\mathbf{A} \otimes \mathbf{B})...(\mathbf{A} \otimes \mathbf{B})}_{l-3} \\
&= (\mathbf{A}^3 \otimes \mathbf{B}^3)\underbrace{(\mathbf{A} \otimes \mathbf{B})...(\mathbf{A} \otimes \mathbf{B})}_{l-3} \\
&= (\mathbf{A}^{(l-1)} \otimes \mathbf{B}^{(l-1)})(\mathbf{A} \otimes \mathbf{B}) \\
&= (\mathbf{A})^l \otimes (\mathbf{B})^l
\end{aligned} \tag{18}$$

The deduction of Eq. (7):

$$\begin{aligned}
\kappa^{(l)}(G_i, S_m) &= \mathbf{1}^{\mathsf{T}}(\mathbf{A}_i \otimes \mathbf{P}_m)^l \mathbf{1} \\
&= \mathbf{1}^{\mathsf{T}}\left[(\mathbf{A}_i)^l \otimes (\mathbf{P}_m)^l\right]\mathbf{1} \\
&= \mathbf{1}^{\mathsf{T}}\left[\left((\mathbf{A}_i)^l\right)^{\mathsf{T}} \otimes (\mathbf{P}_m)^l\right]\mathbf{1} \\
&= \mathbf{1}^{\mathsf{T}}vec\left[(\mathbf{P}_m)^l vec^{-1}(\mathbf{1})(\mathbf{A}_i)^l\right] \\
&= \mathbf{1}^{\mathsf{T}}vec\left[(\mathbf{P}_m)^l \mathbf{E}(\mathbf{A}_i)^l\right] \\
&= \sum_i \sum_j \left[(\mathbf{P}_m)^l \mathbf{E}(\mathbf{A}_i)^l\right]_{i,j},
\end{aligned} \tag{19}$$

where $\mathbf{A}_i \in \mathbb{R}^{n \times n}$ and $\mathbf{P}_m \in \mathbb{R}^{n' \times n'}$ are two symmetric matrices, $\mathbf{1} \in \mathbb{R}^{n' \times n}$ is a column vector and all elements are 1. We get $(\mathbf{A}_i \otimes \mathbf{P}_m)^l = (\mathbf{A}_i)^l \otimes (\mathbf{P}_m)^l$ based on **Proposition 3** and get $\left[\left((\mathbf{A}_i)^l\right)^{\mathsf{T}} \otimes (\mathbf{P}_m)^l\right]\mathbf{1} = vec\left[(\mathbf{P}_m)^l vec^{-1}(\mathbf{1})(\mathbf{A}_i)^l\right]$ based on **Proposition 1**, where $vec^{-1}(\cdot) : \mathbb{R}^{ab} \to \mathbb{R}^{a \times b}$ denotes the inverse vectorization on a vector.

---

**Algorithm 1:** iGAD

---

**Input:**

$\mathcal{T}$: The training set is composed of $N_0$ normal graphs and $N_1$ anomalous graphs;

$\mathcal{Y}_{\mathcal{T}}$: The label set of graphs in $\mathcal{T}$;

$K, L, M, n'$: The maximum hop of neighborhood utilized by the anomalous attribute-aware graph convolution, the maximum length of random walk sequences, the number of anomalous substructures, and the size of anomalous substructures;

$B, E$: The number of batches and epochs.

**Output:** Model Parameters

1 Initializing model parameters (including anomalous substructures $S_m, m \in \{1, ..., M\}$).

2 Calculating the proportion of normal graphs $P(y = 0) = \frac{N_0}{N_0 + N_1}$ in $\mathcal{T}$ and the proportion of anomalous graphs $P(y = 1) = \frac{N_1}{N_0 + N_1}$ in $\mathcal{T}$.

3 **for** $e = 1, ..., E$ **do**

4    **for** $= 1, ..., B$ **do**

5       // In the pseudo-code, we suppose that a batch only contain a graph $G_i$ (i.e., batch_size=1).

6       $\mathbf{h}^{(0)}(u) \leftarrow$ Eq. (4), $\forall u \in G_i$;

7       **for** $k = 1, ..., K$ **do**

8          $\mathbf{h}^{(k)}(u) \leftarrow$ Eq. (5), $\forall u \in G_i$;

9       Obtaining the node representation of node $u$, $\mathbf{h}_u \leftarrow$ Eq. (3), $\forall u \in G_i$;

10      Obtaining the proposed anomalous attribute-aware graph convolution-based graph representation $\mathbf{h}_{G_i}^{Conv}$ of the graph $G_i$, $\mathbf{h}_{G_i}^{Conv} \leftarrow$ Eq. (6);

11      **for** $l = 1, ..., L$ **do**

12         Calculating kernel values, $\kappa^{(l)}(S_m, G_i), m \in \{1, ..., M\} \leftarrow$ Eq. (7);

13      Assembling all kernel values $\kappa^{(l)}(S_m, G_i), l \in \{1, ..., L\}, m \in \{1, ..., M\}$ into a vector, $\mathbf{a}_{G_i} \leftarrow$ Eq. (8);

14      Obtaining the proposed anomalous substructure-aware deep RWK-based graph representation $\mathbf{h}_{G_i}^{ker}$ of graph $G_i$, $\mathbf{h}_{G_i}^{ker} \leftarrow$ Eq. (9);

15      Obtaining the final graph representation $\mathbf{h}_{G_i}$ by concatenating $\mathbf{h}_{G_i}^{Conv}$ and $\mathbf{h}_{G_i}^{Ker}$;

16      Inputing $\mathbf{h}_{G_i}$ in to an MLP and obtaining the prediction result on $G_i$, $y_i \leftarrow$ Eq. (14);

17      Calculating the PMI-based loss function, $\mathcal{L} \leftarrow$ Eq. (13).

---

### A.2

The overall prediction accuracy of a classifier can be approximated as $\frac{1}{N'} \sum_{i=1}^{N'} P_\Theta(y_i|G_i)$. If we do not consider the imbalanced labels, the $y$ that can maximize $p_\Theta(y_i|G_i)$ will be regarded as the label of $G_i$. However, we expect that the graph-level anomaly detection model iGAD can achieve high prediction accuracy on normal and anomalous graphs both in mode test time. Hence, we rewrite the prediction accuracy as:

$$
\begin{aligned}
\text{Accuracy} &\approx \frac{1}{N'} \sum_{i=1}^{N'} P_\Theta(y_i|G_i) \\
&= \frac{1}{N'} \sum_{i=1}^{N'} \frac{P_\Theta(y_i|G_i)}{P(y_i)} P(y_i) \\
&= \sum_{y=0}^{1} P(y) \left( \frac{1}{N'} \sum_{G_i \in \Omega_y} \frac{P_\Theta(y|G_i)}{P(y)} \right),
\end{aligned}
\tag{20}
$$

where $\{(G_i, y_i)\}_{i=1}^{N'}$ is the test/validation set, $\Omega_y = \{G_i | y_i = y, i = 1, ..., N'\}$ is the set of $G_i$ labeled $y$, $y \in \{0, 1\}$. $\frac{1}{N'} \sum_{G_i \in \Omega_y} \frac{P_\Theta(y|G_i)}{P(y)}$ can be regarded as the approximation about the prediction accuracy on all graphs labeled $y$. Therefore, in test time, we need the label that can make $\frac{P_\Theta(y|G_i)}{P(y)}$ achieve its maximum value, that is $y_i^* = \underset{y_i}{\text{argmax}} \frac{P_\Theta(y_i|G_i)}{P(y_i)} = \underset{y_i}{\text{argmax}} \frac{P_\Theta(G_i, y_i)}{P(G_i)P(y_i)} = \underset{y_i}{\text{argmax}} f_{y_i}(G_i; \Theta)$.

**Computational Complexity Analysis.** The computational complexity of the proposed graph convolution and the proposed deep RWK is $\mathcal{O}(|E|)$ and $\mathcal{O}(L(Mn'(n+n')+|E|))$, respectively, where $L$ is the maximum length of random walk sequences, $M$ is the number of anomalous substructures, $n'$ is the size of anomalous substructures. Because that $L, M, n' \ll |E|$, the complexity of deep RWK is $\mathcal{O}(n+|E|)$. In summary, the final computational complexity of iGAD is $\mathcal{O}(n+|E|)$.

# B  Experiments

This section introduces the details of parameter setting (see **B.1**) and additional parameter analysis experiments (see **B.2**).

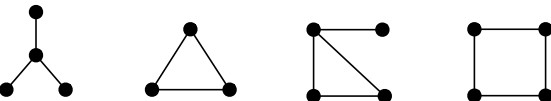

Figure 4: The patterns from left to right are 3-star, triangle, tailed triangle, and 4-cycle graphlets.

## B.1

**Parameter Setting.** Due to the limited number of anomalous graphs, we only divide the training set and validation set for cross-validation experiments. We take the stratified sampling to divide the training and the validation set concerning the division ratio 80% to 20%. Stratified sampling ensures that the training set and validation set have a consistent ratio between normal and anomalous graphs. We take the average of validation results across 5 folds, and select a single epoch that can achieve the maximum average *recall*. The standard deviation is computed at the selected epoch over the 5 folds.

We use the learning rate (0.001), batch size (128), the maximum hop of neighborhoods ($K = 2$), the number of anomalous substructures ($M = 5$), the size of anomalous substructures ($n' = 10$), the max random walk sequence length ($L = 6$), the graph embedding dimension output by graph convolution (32), the graph embedding dimension output by deep RWK (16), and the final graph embedding dimension (32), neighbor_aggregation_type (sum), readout (mean), and the optimizer (Adam [2]). iGAD-2 utilizes pre-defined anomalous substructures, it has the same parameter setting as iGAD except for $M$ and $n'$. In addition, the parameter setting on baseline models is listed below:

- GCN & GCN-GAD: {learning_rate: 0.001; batch_ size: 128; num_convolution_layer: 2; embedding_dimension: 32; graph_pooling_type: sum; optimizer: Adam}
- GIN & GIN-GAD: {learning_rate: 0.01; batch_ size: 128; num_layers: 5; num_mlp_layers: 2; embedding_dimension: 32; final_dropout: 0.5; neighbor_pooling_type: sum; graph_pooling_type: sum; optimizer: Adam}
- DGCNN & DGCNN-GAD: {learning rate: 0.0001; sort_pooling_k: 6; batch_size: 128; embedding_ dimension: 128; convolution_dimension: 32-32-32-1; optimizer: Adam}
- SOPool & SOPool-GAD: {learning rate: 0.01; num_layers: 5; num_mlp_layers: 2; hidden_dimension: 64; final_dropout: 0.5; neighbor_pooling_type: sum; graph_pooling_type: sum; optimizer: Adam}
- RWGNN & RWGNN-GAD: {learning rate: 0.01; dropout: 0.2; batch_size: 128; num_hidden_graphs: 5; size_hidden_graphs: 5; max_step: 2; hidden_dimension: 32; penultimate_dimension: 32; optimizer: Adam}

**Implementation.** All algorithms are implemented on Python 3.7.4, CUDA 10.1, cuDNN 7.6.5, 1 NVIDIA Volta GPU, 384GB RAM, 2×24-core Intel Xeon Platinum 8268 Linux desktop. For GCN, GIN, DGCNN, SOPool, and RWGNN, we use their official implementation on GitHub. For GCN-GAD, GIN-GAD, DGCNN-GAD, SOPool-GAD, and RWGNN-GAD, we replace the cross-entropy loss in the official implementation with the proposed PMI-based loss function.

**Evaluation Metrics.** We utilize *AUC*, *Recall* (average="macro"), *Recall(A)*, and *F-score* (average="macro") as evaluation metrics. *AUC* denotes the probability that the ranking of a randomly selected anomalous graph is higher than that of a normal graph. *Recall* denotes the average value of the proportion of correctly predicted normal graphs in all normal graphs and the proportion of correctly predicted anomalous graphs in all anomalous graphs. *Recall(A)* denotes the proportion of correctly predicted anomalous graphs in all anomalous graphs. *F-score* comprehensively considers *Recall* and *Precision*, where *Precision* indicates the average value of the proportion of true normal graphs in the graphs predicted to be normal and the proportion of true anomalous graphs in the graphs predicted to be anomalous. *Recall* and *Precision* are a pair of negatively correlated indicators. In this paper, we choose *Recall* because iGAD is supposed to recall as many anti-cancer molecules as possible.

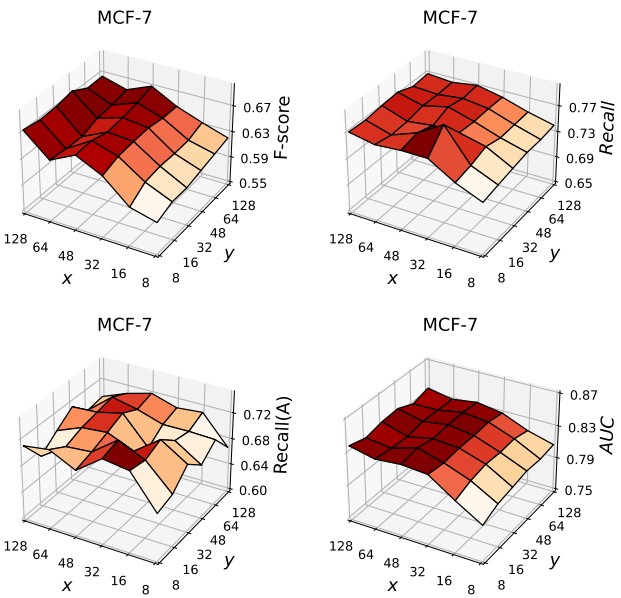

Figure 5: The parameter analysis result of iGAD on the graph dataset MCF-7. The $x$-axis and $y$-axis denote the embedding dimension output by the proposed graph convolution and deep RWK, respectively. The $z$-axis denotes the model performance *w.r.t.*, *F-score*, *Recall*, *Recall(A)*, and *AUC*.

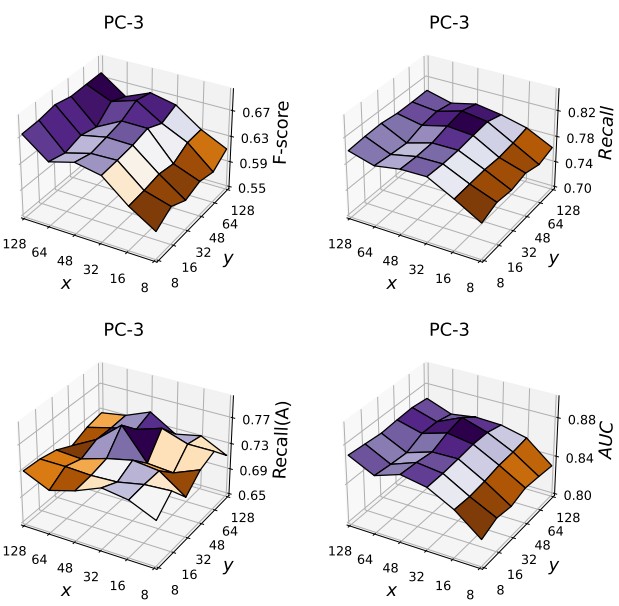

Figure 6: The parameter analysis result of iGAD on the graph dataset PC-3.

**B.2**

**Parameter Analysis.** Figures 5 and 6 demonstrate the influence of embedding dimension output by the proposed graph convolution and the proposed deep RWK on the performance of iGAD on MCF-7 and PC-3, respectively. iGAD can achieve good performance under different representation dimension settings, which verifies that iGAD is not sensitive to the embedding dimension. In addition, we can also observe that the representation dimension output by the proposed graph convolution has a greater impact on the performance of iGAD than that of the proposed deep RWK.