# OpenReview forum: "Dual-discriminative Graph Neural Network for Imbalanced Graph-level Anomaly Detection"
_NeurIPS.cc/2022/Conference — NeurIPS 2022 Accept_

### Official Review · Reviewer_yeLE · 2022-06-15

**Rating:** 6
**Confidence:** 5
**Soundness:** 3 good
**Presentation:** 3 good
**Contribution:** 3 good

**Summary:**

This paper  is applied in the imbalance problem of graph-level anomaly detection, they propose a dual-discriminative graph neural network for graph-level anomaly detection. The first module is a novel anomalous attribute-aware graph convolution, which separates self representation and neighborhood-representation when representing a node in a graph. The second module is  an anomalous graph
substructure-aware deep Random Walk Kernel to  encode anomalous structural information into graph representations.  Further,  they propose a Point Mutual Information-based loss function to address the imbalance nature of anomaly problem.

**Questions:**

1. I think anomaly detection task  is originally a problem of label imbalance, and the author should not repeatedly emphasize this.

2. This paper has limited application value. Due to there are very few anomaly detection labels, although supervised learning can also solve this problem, it is difficult to obtain labels of anomaly patterns in practical applications. The author should consider methods that do not rely on label information as much as possible, such as semi-supervised and unsupervised learning, e.g.

3. The authors should compare with the most advanced methods for graph-level AD task, instead of GCN, GIN, SOPool, e.g.

For example:

1): Ma R, Pang G, Chen L, et al. Deep Graph-level Anomaly Detection by Glocal Knowledge Distillation[C]//Proceedings of the Fifteenth ACM International Conference on Web Search and Data Mining. 2022: 704-714.

2): Qiu C, Kloft M, Mandt S, et al. Raising the Bar in Graph-level Anomaly Detection[J]. arXiv preprint arXiv:2205.13845, 2022.


**Limitations:**

Please see Questions part

**Strengths And Weaknesses:**

Strengths:

1. Experiment are done on extensive real-world graph datasets and  evaluations are sufficient, moreover achieve great performance improvement.

2. The idea is relatively novel， which couples  attribute-aware and  substructure-aware knowledge for graph-level AD problem.

Weaknesses:

1. The paper is not organized well, especially in the section of  Introduction and Related Work.

2. Figure 1 is not intuitive enough to understand the intention of the authors.

---

> ### Author Response · Authors · 2022-08-02
> **Author Response**
>
> We are grateful for your constructive reviews.
>
> Thanks very much for your attention to the paper’s organization. We are glad that the organization has got some acknowledgment in the peer review of NeurIPS, we believe that it can be better once provided with more details. In addition, we have revised Figure 1 and its caption to make this figure to be more intuitive. Please refer to the anonymous link:
> https://github.com/iGADrebuttal/9811_NeurIPS_rebuttal
>
> The response to the reviewer’s questions are as follows:
>
> **Q1:** Anomaly detection task is originally a problem of label imbalance, authors should not repeatedly emphasize this.
>
> **R1:**  We emphasize this as many papers targeting anomaly detection have overlooked this critical problem. For example, we regret to find that the two latest graph-level anomaly detection models [1,2] cannot work on data with highly imbalanced label distribution. However, such data distribution is very common for anomaly detection. The detailed experiment results can be found in the response **R3**. [2] follows one-class classification that has achieved empirical success in anomaly detection. Perhaps the low performance of [2] can also be attributed to the blind use of graph convolution. We expect that iGAD can encourage graph-level anomaly detection researchers to pay more attention to label imbalance.
>
> **Q2:** This paper has limited application value. Due to there are very few anomaly detection labels, although supervised learning can also solve this problem, it is difficult to obtain labels of anomaly patterns in practical applications. The author should consider methods that do not rely on label information as much as possible, such as semi-supervised and unsupervised learning, e.g.
>
> **R2:** Imbalanced graph-level anomaly detection is a research topic worthy of study, but we also agree that unsupervised learning is more economical for anomaly detection. Currently, there are still some challenges in unsupervised anomaly detection. For example, 1) how to choose an appropriate threshold for the learned abnormal score to divide normal and abnormal samples; 2) how to improve the relatively poor model performance [1,2]. We would like to develop unsupervised graph-level anomaly detection models in the future.
>
> **Q3:** The authors should compare with the most advanced methods for graph-level AD task ([1] Ma et al., WSDM 2022 and [2] Qiu et al., Arxiv 2022), instead of GCN, GIN, SOPool, e.g.
>
> **R3:** We are grateful to the reviewer for listing these two latest algorithms. We have run their codes on datasets with highly imbalanced labels used by iGAD (i.e., MCF-7, PC-3, MOLT-4, SW-620). Unfortunately, our findings show that [1,2] are not suitable for graph-level anomaly detection. Because that: 1) they adopt the graph convolution that is not applicable for detecting anomalous graphs. 2) they do not consider the imbalance labels. The details are as follows:
>
> 1. [1] can only use AUC as the evaluation metric. When running [1] on the dataset used by iGAD with highly imbalanced labels, from the first epoch to the last, almost no improvement w.r.t. AUC can be seen. [1] uses node degrees as node attributes for graphs without explicit node features. Worrying that node degrees are not informational enough, we revise the code and adopt one-hot representations of node labels as node attributes, but the performance is still poor. When running [1] on the dataset originally used by [1], we find that [1] can indeed achieve a good performance on binary data with balanced labels, such as AIDS. However, as shown in Table 1 in [1], the performance of [1] on data with multi labels or with real anomalies is significantly reduced, compared with its performance on balanced binary data, e.g., AIDS.
>
> 2. [2] can be evaluated by metrics Recall, Recall(A), and F-score. However, [2] only reports its model performance w.r.t. AUC. The below table shows that [2] can hardly recall any anomalous graphs.
>
>
> |Data(proportion of abnormal graphs)|Metric|iGAD(ours)|[1]|[2]|
> |----|----|----|----|----|
> |MCF-7(8.26%)|AUC|83.22+-0.64|61.43+-1.26|64.92+-1.92|
> ||Recall|76.77+-0.47|/|47.19+-0.01|
> ||Recall(A)|73.38+-3.59|/|0.30+-0.48|
> ||F-score|64.70+-2.58|/|52.88+-1.59|
> |PC-3(5.7%)|AUC|86.04+-1.14|64.79+-1.22|68.42+-1.73|
> ||Recall|79.59+-0.41|/|49.08+-0.02|
> ||Recall(A)|75.69+-1.64|/|1.91+-0.05|
> ||F-score|63.50+-0.73|/|54.46+-1.87|
> |MOLT-4(7.9%)|AUC|83.59+-1.07|61.43+-1.26|57.43+-2.38|
> ||Recall|76.82+-0.82|/|46.08+-0.01|
> ||Recall(A)|72.10+-2.47|/|0.10+-0.15|
> ||F-score|63.30+-1.17|/|53.06+-1.38|
> |SW-620(5.95%)|AUC|85.82+-0.69|64.14+-0.92|67.69+-0.02|
> ||Recall|79.64+-0.83|/|48.03+-0.01|
> ||Recall(A)|74.27+-2.99|/|0.04+-0.13|
> ||F-score|63.68+-1.56|/|53.55+-0.03|
>
> ## Reference
>
> [1] Ma et al. Deep Graph-level Anomaly Detection by Glocal Knowledge Distillation. WSDM, 2022. (from the reviewer yeLE)
>
> [2] Qiu et al. Raising the Bar in Graph-level Anomaly Detection. arXiv, 2022. (from the reviewer yeLE)

---

> > ### Author Response · Authors · 2022-08-09
> > **Re: Author Response**
> >
> > Dear Reviewer yeLE,
> >
> > We sincerely appreciate your time for the valuable review comments. And we also sincerely hope that our response to your comments, especially our clarifications and additional comparison experiments per your suggestion, can help address your concerns. Please do not hesitate to let us know if you have anything further unclear or confused. We are more than happy to address your concerns.
> >
> >
> > Best Regards,
> >
> > Authors

---

### Official Review · Reviewer_eQtL · 2022-07-05

**Rating:** 8
**Confidence:** 5
**Soundness:** 4 excellent
**Presentation:** 3 good
**Contribution:** 4 excellent

**Summary:**

The paper proposes iGAD, a supervised anomaly detection model for distinguishing anomalous graphs from normal graphs. This paper identifies anomalous attributes and/or anomalous substructures as the anomaly notions of graphs, and also notices the imbalance nature of the anomaly problem. The authors employed three designs to achieve the graph-level anomaly detection model iGAD: a. Let the attributes of each node evolve individually in the graph convolution process to emphasize the anomalous attributes; b. Considering the defect of message passing-based GNNs in learning structural information, iGAD employs a deep random walk kernel that utilizes discriminative substructure learning and structure comparison to explore anomalous substructures; c. A new loss function was proposed to enable the proposed model to fit the minority class samples (i.e., anomalous graphs) well. Experimentally, iGAD shows strong performances in classifying anomalous and normal graphs on four graph datasets.

**Questions:**

Some questions to help authors further improve their research:
1. The authors put forward a simple and general solution for learning the imbalanced data distribution. But different downstream application tasks have different tolerance for the false-positive rate and the false-negative rate. Considering it may help improve the applicability of iGAD.
2. The datasets used for model verification are all molecular graphs. Why did not the author use the graph dataset from other application domains and have multi labels and specify one of the labels as anomalous and others as normal to conduct experiments?
3. I suggest authors give more detailed explanations on why the cross-entropy loss cannot work on imbalanced data distribution in the main content or the appendix.

**Limitations:**

The authors clearly call out the limitation of their work.

**Strengths And Weaknesses:**

Originality: This paper proposes an interesting method to detect anomalous graphs in a graph dataset. It is a topic neglected by the anomaly detection and graph mining communities. The method comprehensively considers the anomalous notions in attributes and substructures of graphs, and the imbalance nature of the graph dataset with anomaly graph samples.
Quality: The proposed method is technically sound, and the authors have basically made experimental verification of the proposed designs including the message passing graph convolution, deep RWK, and the PMI-based loss function.
Clarity: This paper has good organization and writing. The author gives detailed preliminaries corresponding to the method to help readers understand the proposed algorithm. In terms of experiments, the author gives the details of parameter settings for readers to reproduce the experiments.
Significance: Few deep graph learning practitioners consider how to detect anomaly graphs. Most graph anomaly detection work focuses on detecting anomaly nodes in a single graph. This paper extends the graph anomaly detection to the whole graph level. Although the anomalous information/notions of a graph may reflect in other aspects in addition to its node attributes and graph structure (authors also admitted this viewpoint in the discussion about the limitations of this work). The learned discriminative substructures provide explainable information for anomalous graph detection.  In addition, the PMI-based loss proposed in this paper has a good generalization, it can be extended to other imbalanced classification problems easily. Overall, this paper can be a good starting point for graph-level anomaly detection topic.

---

> ### Author Response · Authors · 2022-08-02
> **Author Response**
>
> We greatly thank you for the constructive comments and positive reviews. Our response is as follows:
>
> **Q1:** The authors put forward a simple and general solution for learning the imbalanced data distribution. But different downstream application tasks have different tolerance for the false-positive rate and the false-negative rate. Considering it may help improve the applicability of iGAD.
>
> **R1:** The most common applications of graph-level anomaly detection include abnormal brain network detection, anti-cancer molecules detection, and detecting proteins with special functions. The above tasks require graph-level anomaly detection models can achieve a relatively low false-negative rate or the high recall on anomalous graphs. Our experiments have proved that iGAD can achieved good performance in recalling anomalous graphs. Building graph-level anomaly detection models that can adapt to various application scenarios having different tolerance for FPR and FNR is exciting. Thank you very much for highlighting the interesting future directions for us.
>
> **Q2:** The datasets used for model verification are all molecular graphs. Why did not the author use the graph dataset from other application domains and have multi labels and specify one of the labels as anomalous and others as normal to conduct experiments?
>
> **R2:** This way the reviewer mentioned can indeed create more dataset. Unfortunately, it is unreasonable to use such datasets to train a graph-level anomaly detection model. An ideal graph-level anomaly detection model distinguishes anomalous graphs from normal graphs by identifying the anomalous and normal features. Obtaining dataset in the way the reviewer mentioned cannot ensure that:
>
> 1) the graphs regarded as anomalous graphs have some features that can work as anomalous features;
>
> 2) the graphs regarded as normal graphs have common features that can be used as normal features.
>
> Under such concerns, we cannot guarantee that the trained model is an indeed anomaly detection model.
>
> **Q3:** I suggest authors give more detailed explanations on why the cross-entropy loss cannot work on imbalanced data distribution in the main content or the appendix.
>
> **R3:** Thanks for this suggestion. It is a good suggestion to help readers understand our paper deeply.
> Minimizing the cross-entropy loss is equal to improving the prediction accuracy (ACC) of a model. ACC = (TP+TN)/(TP+TN+FP+FN). However, accuracy is not a reasonable evaluation metric for anomaly detection. An anomaly detection model that correctly predicts all normal graphs, but incorrectly predicts all anomalous graphs can still achieve high accuracy. For example, in the dataset SW-620 with 5.95% anomalous graphs (see Table 1 in our submission), a graph-level anomaly detection model that predicts all graphs as normal graphs can also achieve 94.5% accuracy. We would like to add the above discussion to the appendix.

---

> > ### Comment · Reviewer_eQtL · 2022-08-08
> > **More comments**
> >
> > Thanks for the compelling response and clarification. My questions are addressed carefully. Yes, more discussions on the loss function can provide further insight into the research work and enhance the technical contribution. I still think that imbalanced graph-level anomaly detection is a topic of great interest, and the paper contributes well to this topic.

---

> > > ### Author Response · Authors · 2022-08-09
> > > **Thank You!**
> > >
> > > We sincerely appreciate your time and positive review for recognizing the significance of our contributions, the interesting research work, the quality of our method, organization and writing. Thank you again.

---

### Official Review · Reviewer_653N · 2022-07-10

**Rating:** 5
**Confidence:** 4
**Soundness:** 3 good
**Presentation:** 3 good
**Contribution:** 2 fair

**Summary:**

This paper focuses on a problem of graph-level anomaly detection that distinguish anomalous graphs from a set of graphs. For one graph, it could be considered as anomaly due to the anomalous attributes of certain nodes and the anomalous substructures. It is argued in the paper that generally GNNs will aggregate the information of neighbors resulting the relatively poor representation for anomaly graph detection. Thus, the authors investigate an anomalous attribute-aware graph convolution and anomalous substructure-ware deep RWK to detect the anomalies. Specifically, in anomalous attribute-aware graph convolution, the node representation and the representation of its neighbor will be separately encoded instead of merging together as GCN. To detect the anomalous substructure, the paper proposes a way to compare the random walk paths of graphs and the anomalous substructure. The anomalous substructure's adjacency matrix is learned in an end-to-end manner.  A PMI-based loss function is used to balance the contributions of normal graphs and anomalous graphs. Experiments are conducted several large-scale real-world to evaluate the effectiveness of the proposed method.

**Questions:**

1. It is highly suggested to clarify  the technical contribution of the proposed iGAD and have more discussions about the differences of iGAD with existing anomaly detection methods on graphs.
2. It would be more convincing if experiments in comparing the potential baselines mentioned above in graph-level node classification can be added.

**Limitations:**

Please refer to the weaknesses for details.

**Strengths And Weaknesses:**

**Strengths:**
1. The graph-level anomaly detection is an important problem to be solved. And the existing work generally focus on node-level anomaly detection in graph. The proposed iGAD can give some insights for the community.
2. The challenges of  graph-level anomaly detection are discussed. And the proposed mechanisms are convincing in detecting the anomalous attributes and anomalous sub-structures.
3. The paper is well organized and easy to follow.
4. Based on the current results, the iGAD seems promising in detecting anomalous graphs on the real-world datasets.

**Weaknesses***
1. Though the iGAD focuses on a novel problem of graph-level anomaly detection, there are concerns of technical contributions of the proposed method.
    * Firstly, as node-level anomaly detection also faces the anomalous nodes and anomalous graphs, many methods in node anomaly detection share similar high-level idea with iGAD. For example, the anomaly scores based on the aspects of attribute matrix and adjacency matrix are separately considered in [1,2] for node anomaly detection. Thus, it is suggested to have more discussions about how the iGAD can address the unique challenges of graph-level anomaly detection.
    * Secondly, it is argued that GNNs will aggregate the information of neighbors resulting the relatively poor representation for anomaly graph detection. To address this challenge, the graph convolution in iGAD separates the encoding on a node itself from its neighborhood when representing nodes. This design also have been widely known in GraphSage and H2GCN [3] to avoid mixing dissimilar nodes in H2GCN.

2. Some important references and potential baselines are not covered. There are already works [4,5] noticed that the GNNs may not work well for anomaly detection because of the aggregation mechanism. They have proposed some special aggregation mechanism for anomaly detection. Though they are originally designed for node-level anomalies, it would be easy to extend for graph-level anomaly by adding readout function. I would suggest to compare with these state-of-the-art anomaly detection methods.

References
[1] Bandyopadhyay, Sambaran, Saley Vishal Vivek, and M. N. Murty. "Outlier resistant unsupervised deep architectures for attributed network embedding." Proceedings of the 13th international conference on web search and data mining. 2020.
[2]  Kumagai, Atsutoshi, Tomoharu Iwata, and Yasuhiro Fujiwara. "Semi-supervised anomaly detection on attributed graphs." 2021 International Joint Conference on Neural Networks (IJCNN). IEEE, 2021.
[3] Zhu, Jiong, et al. "Beyond homophily in graph neural networks: Current limitations and effective designs." Advances in Neural Information Processing Systems 33 (2020): 7793-7804.
[4] Kumagai, Atsutoshi, Tomoharu Iwata, and Yasuhiro Fujiwara. "Semi-supervised anomaly detection on attributed graphs." 2021 International Joint Conference on Neural Networks (IJCNN). IEEE, 2021.
[5] Dou, Yingtong, et al. "Enhancing graph neural network-based fraud detectors against camouflaged fraudsters." Proceedings of the 29th ACM International Conference on Information & Knowledge Management. 2020.

---

> ### Author Response · Authors · 2022-08-02
> **Author Response 1/2**
>
> Thanks for your reviews and the references [1-5] you list for us.
>
> P.S. we notice that the references [2] and [4] are the same.
>
> **Q1:** Though the iGAD focuses on a novel problem of graph-level anomaly detection, there are concerns of technical contributions of the proposed method.
>
> **R1:** We have read your concerns on iGAD’s technical contributions carefully. These concerns can be addressed by highlighting the significant differences between node-level and graph-level anomaly detection. The detailed explanation and response are as follows:
>
> **Q1.1:** Firstly, as node-level anomaly detection also faces the anomalous nodes and anomalous graphs, many methods in node anomaly detection share similar high-level idea with iGAD. For example, the anomaly scores based on the aspects of attribute matrix and adjacency matrix are separately considered in [1,2] for node anomaly detection. Thus, it is suggested to have more discussions about how the iGAD can address the unique challenges of graph-level anomaly detection.
>
> **R1.1:** Graph anomaly detection includes: 1) node-level anomaly detection which detects abnormal/anomalous nodes in a graph; 2) graph-level anomaly detection which identifies abnormal/anomalous graphs from a collection of graphs.
>
> The discussion about [1,2]:
>
> P.S. [2] does not separately consider graph attribute and topology in calculating anomaly scores, it only employs GCN to encode nodes, as shown in Figure 1 in [2].
>
> Although node-level anomaly detection models [1,2] identify abnormal nodes by analyzing graph attributes and topology, they cannot be used to detect abnormal graphs mainly because 1) node-level and graph-level anomaly detection have different definitions of the abnormal node attributes; 2) a graph-level anomaly detection model should be able to explore all possible substructures, but [1,2] cannot.
>
> The details about the unique challenges of graph-level anomaly detection and how iGAD can address these challenges are as follow:
>
> 1. In most node-level anomaly detection models [1,5,6,7,8], nodes that violate homophily are anomalies, i.e., a node with different attributes/labels from its neighbors (or node within the same community) is abnormal. This opinion is theoretically proved by [7]. Most node-level anomaly detection models identify anomalies by analyzing node attributes and structures separately or jointly [1,2,5,6,7,8]. However, the structure here is generally limited to the node neighborhood.
>
> 2. **In graph-level anomaly detection, anomalous substructures are not limited to the neighborhood of a node. The anomalous substructure information could be all possible substructures (unique challenge #1)**. However, GNNs focus on the aggregation of the neighbor-structure information. Hence, we introduce the deep RWK to iGAD to identify anomalous substructures (how we address challenge #1). Deep RWK enables iGAD to explore substructures more substructures, instead of being limited in the node neighborhood. **In addition, detecting nodes that violate homophily is very common in node-level anomaly detection [5,6,7,8], but it is not helpful for graph-level anomaly detection. We cannot say that a graph having nodes that violate homophily is abnormal (unique challenge #2)**. We propose the anomalous attribute-aware graph convolution to expose node attributes that can be used to distinguish anomalous graphs from normal graphs (how we address challenge #2).
>
> **Q1.2:** Secondly, it is argued that GNNs will aggregate the information of neighbors resulting the relatively poor representation for anomaly graph detection. To address this challenge, the graph convolution in iGAD separates the encoding on a node itself from its neighborhood when representing nodes. This design also has been widely known in GraphSage and H2GCN [3] to avoid mixing dissimilar nodes in H2GCN.
>
> **R1.2:** The aggregation mechanism of GraphSage and H2GCN are not suitable for graph-level anomaly detection, since that:
>
> 1. GraphSage iteratively performs feature transformations on the vector that concatenates a center node and its neighbors. This design will break the individual evolution of center nodes in the graph convolution process and make anomalous node attributes difficult to be identified.
>
> 2. H2GCN is easy to suffer from memory overflow [9], as shown in Table 3 in [9]. H2GCN must form the squared adjacency $\mathbf A^2$ since it follows the insight that a node has similar attributes to its two-hop neighbors. Compared with H2GCN, iGAD adopts a more efficient and lightweight design to encode nodes.

---

> ### Author Response · Authors · 2022-08-02
> **Author Response 2/2**
>
> **Q2:** Some important references and potential baselines are not covered. There are already works [4,5] noticed that the GNNs may not work well for anomaly detection because of the aggregation mechanism. They have proposed some special aggregation mechanism for anomaly detection. Though they are originally designed for node-level anomalies, it would be easy to extend for graph-level anomaly by adding readout function. I would suggest to compare with these state-of-the-art anomaly detection methods.
>
> **R2:** [4, 5] are proposed for node-level anomaly detection.
>
> [4] adopts the graph convolution of GCN directly, and it does not propose any special aggregation mechanism. GCN is one of our baselines. Please see Table 2 in our submission for the experiment results. In addition, iGAD-1 is a variant of iGAD, it only uses our proposed graph convolution (i.e., the anomalous attribute-aware graph convolution) to detect anomalous graphs. As shown in Table 2 in our submission, iGAD-1 greatly outperforms baselines, which can verify that our proposed graph convolution is effective.
>
> [5] proposed a similarity-aware neighbor selector that only aggregates neighbors which has similar attributes and same labels with the center node to the center node. In essence, [5] identifies nodes that violate homophily as abnormal nodes. However, the graph-level anomaly detection task does not regard graphs that have nodes violating homophily as abnormal graphs. Therefore, filtering dissimilar neighbors for center nodes like [5] is useless for graph-level anomaly detection and can cause a lot of information loss.
>
> Overall, node-level and graph-level anomaly detection models need to consider special aggregation mechanisms both, but they have different definitions of abnormal/anomalous node attributes. Graph-level anomaly detection models do not consider that a graph having nodes that violate homophily is abnormal, but most node-level anomaly detection models [5,6,7,8] are proposed for detecting nodes violating homophily. It is unfair to employ node-level anomaly detection models as the baselines to verify that a graph-level anomaly detection model is effective.
>
> **Q3:** It is highly suggested to clarify the technical contribution of the proposed iGAD and have more discussions about the differences of iGAD with existing anomaly detection methods on graphs.
>
> **R3:** Thank you for this suggestion. Please refer to the response **R1.1**.
>
> **Q4:** It would be more convincing if experiments in comparing the potential baselines mentioned above in graph-level node classification can be added.
>
> **R4:** Thank you for this suggestion. Please refer to our responses **R2** and **R1.1**.
>
>
> ## Reference:
>
> [1] Bandyopadhyay, Sambaran, et al. Outlier resistant unsupervised deep architectures for attributed network embedding. WSDM, 2020.  (from the reviewer 653N)
>
> [2] Kumagai, Atsutoshi, et al. Semi-supervised anomaly detection on attributed graphs. IJCNN, 2021. (from the reviewer 653N)
>
> [3] Zhu, Jiong, et al. Beyond homophily in graph neural networks: Current limitations and effective designs. NeurIPS, 2020. (from the reviewer 653N)
>
> [4] Kumagai, Atsutoshi, et al. Semi-supervised anomaly detection on attributed graphs. IJCNN, 2021. (from the reviewer 653N, same as [2])
>
> [5] Dou, Yingtong, et al. Enhancing graph neural network-based fraud detectors against camouflaged fraudsters. CIKM, 2020. (from the reviewer 653N, same as [2])
>
> [6] Liu, Yang, et al. Pick and choose: a GNN-based imbalanced learning approach for fraud detection. WWW, 2021.
>
> [7] Tang, Jianheng, et al. Rethinking Graph Neural Networks for Anomaly Detection. ICML, 2022.
>
> [8] Ma, Xiaoxiao, et al. A comprehensive survey on graph anomaly detection with deep learning. IEEE Trans Knowl Data Eng, 2021.
>
> [9] Lim, Derek, et al. Largescale learning on non-homophilous graphs: new benchmarks and strong simple methods. NeurIPS, 2021.

---

> ### Author Response · Authors · 2022-08-09
> **Re: Author Response 1/2 & 2/2**
>
> Dear Reviewer 653N:
>
> We sincerely appreciate your time and support of the importance of our research problem, our convincing algorithm iGAD, and the paper organization, particularly for the promising algorithm performance in real-world datasets. And we also sincerely hope that our response can help address your concerns and ensure no scope for misunderstanding in the aspect pointed out by you, especially on the significant differences between node-level and graph-level anomaly detection.
>
> Please allow us to emphasize the significant differences between these two topics:
>
> 1. Most node-level anomaly detection methods regard nodes that violate homophily as anomalies. By contrast, for graph-level anomaly detection, the graph having nodes that violate homophily is not always anomalous and could be a normal graph. For example, for molecular graphs, it is very common that two different atoms (i.e., nodes with different attributes) are connected, regardless of the properties of these molecules.
>
> 2. The majority of node-level anomaly detection algorithms focus on the neighborhood structure, since they aim to detect nodes that betray homophily. However, in graph-level anomaly detection, anomalous structures could be all possible substructures.
>
> We have discussed the advantages of the proposed anomalous attribute-aware graph convolution compared with GraphSage and G2GCN.
>
> We have already clarified the reasons why the node-level anomaly detection methods equipped with the readout function cannot be the appropriate baseline for graph-level anomaly detection.
>
> As the discussion time between reviewers and authors is close to the end, we would like to see if you have anything further unclear or confused. We are more than happy to answer your questions to clear up any remaining misunderstandings regarding our research.
>
>
> Best Regards,
>
> Authors

---

> > ### Comment · Reviewer_653N · 2022-08-09
> > **Thanks for the response**
> >
> > I appreciate the clarifications from the authors.
> >
> > I agree with the authors that in some situations the node-level anomalies are different from the graph anomalies. However, in many cases, the node anomalies and graph anomalies share very common patterns. For instance, it is likely that a graph with several anomaly nodes are regarded as anomalies. In addition, a lot of existing node-level anomaly detection methods do consider from both node features and adjacency matrix to evaluate the anomaly score, which can be easily extended to the graph anomaly detection by averaging the node anomaly scores or adding a READOUT function. So I believe it is necessary to compare with such highly related SOA node-level anomaly detection methods to demonstrate the superiority of the method in graph-level anomaly. Thus, I will keep the score unchanged.

---

> > > ### Author Response · Authors · 2022-08-10
> > > **Experiment Results**
> > >
> > > Dear Reviewer 653N,
> > >
> > > Thanks for your response. We have run the node-level anomaly detection algorithm DOMINANT on datasets MCF-7, PC-3, MOLT-4, and SW-620. DOMINANT is a representative and well-used node-level anomaly detection algorithm, which learns the anomaly score of a node based on node features and adjacency matrix.
> > > Following your suggestion, to enable DOMINANT to perform graph-level anomaly detection, we sum or average all node anomaly scores in a graph to achieve DOMINANT (sum) and DOMINANT (mean), respectively. The experiment results are as follows:
> > >
> > > |AUC|DOMINANT(mean)|DOMINANT(sum)|iGAD(ours)|
> > > |---|:--:|:--:|:--:|
> > > |**MCF-7**|49.43|63.66|83.22|
> > > |**PC-3**|51.98| 67.32|86.04|
> > > |**MOLT-4**|50.88|63.15|83.59|
> > > |**SW-620**|50.84|65.23|85.82|
> > >
> > > The experiment results further demonstrate the superiority of our proposed graph-level anomaly detection method iGAD.
> > >
> > > Combing a node-level anomaly detection method with Readout to detect anomalous graphs are confronted with limitations, since:
> > > 1. Node-level anomaly detection methods identify anomaly nodes from a micro perspective (e.g., neighborhood structure). Graph-level anomaly detection methods learn graphs and nodes from a macro perspective.
> > > 2. The forms of graph-level anomalies are more diversified than those of node-level anomalies. Employing node-level anomaly detection methods to detect graph-level anomalies could capture some anomalies, but only a small part.
> > >
> > > **[DOMINANT]** Ding, Kaize, et al. "Deep anomaly detection on attributed networks." Proceedings of the 2019 SIAM International Conference on Data Mining, 2019.
> > >
> > > Best Regards,
> > >
> > > Authors

---

### Official Review · Reviewer_qTND · 2022-07-18

**Rating:** 7
**Confidence:** 4
**Soundness:** 3 good
**Presentation:** 3 good
**Contribution:** 3 good

**Summary:**

In this paper the problem of graph level anomaly detection is studied, for imbalanced data, and a dual discriminative graph neural network is proposed, called iGAD. The proposed method focus on two main aspects of anomaly detection in graphs; (i) anomalous nodes attributes, and (ii) anomalous graph structures. For this reason, the method uses (i) anomalous attribute-aware graph convolution and (ii) anomalous substructure-aware deep random walk graph kernels. To deal with the imbalance that appears usually in anomaly detection scenarios, a point wise mutual information (PMI) loss function is used between the input graphs and their anomalous/normal attributes. A series of experiments have been conducted to study the performance of iGAD against other state of the art comparison methods in four real world chemical compound datasets and investigate a couple of research questions.

**Questions:**

- In Table 2: Why the authors do not give also the results for the Precision metric?

- In Table 2: Are all these results tested for statistical significance? It is important to mention the statistical test and score for these results, as the 20% of the datasets that is being tested, is a small dataset.

- Did the authors try bigger size of random walks (> 8 nodes)? To better understand this experiment, it would be useful to add in the Table 1 with the dataset statistics: the number of connected components in each graph and the avg. min. max. size of the connected components in each graph.

**Limitations:**

The authors did not address the potential negative societal impact of the work, however there is no potential negative impact.
The authors did address some of the limitations of the work and they leave them for future work.

**Strengths And Weaknesses:**

Strengths:
1) The studied problem and the proposed method are very interesting to the research community. This work combines two main features in anomalous graphs with a novel loss function. The paper is well-written, well-structured and easy to follow.

2) The methodology is sound and the proposed components could be used in different use cases.

3) The authors did a good job in the related work section, organizing and covering some of the related literature to give the reader more motivation. It can be improved if the authors mention the contribution of their method after every limitation shoed in the related works.

4) Experimental setup is very thorough and to the detail. The authors provide six research questions, and they show a detailed comparison with five comparison methods, five variants of the comparison methods and three variations of their method to prove the effectiveness of all the components combined.

Weaknesses:
1) The novelty of this work is limited, as most of the proposed methodology is either already published, or with small modifications from published methodology.

Minor comments:
1) In Figure 1, the step III is confusing, as it appears above step II. In step II, the light blue nodes (that show the random walks) is not clear to which paths they refer to from the graph G_S. I would suggest adding the node id to help the reader follow the thought.

2) In Table 2, in MCF-7 dataset, the best result is not highlighted (iGAD-2).

3) It would help if the authors repeat what M and n’ refer to in the caption of Figure 3.

---

> ### Author Response · Authors · 2022-08-02
> **Author Response**
>
> We thank the reviewer for constructive and insightful reviews. According to your detailed suggestions, three minor comments are all addressed. Please refer to the anonymous link:
> https://github.com/iGADrebuttal/9811_NeurIPS_rebuttal
>
> The response to the reviewer’s questions are as follows:
>
> **Q1:** The novelty of this work is limited, as most of the proposed methodology is either already published, or with small modifications from published methodology.
>
> **R1:** Thanks for your attention to iGAD’s novelties. The novelties of iGAD are: 1) iGAD is proposed for the graph-level anomaly detection task that has been overlooked by most researchers in graph learning and anomaly detection communities. iGAD is effective and can give some insights to researchers who are interested in graph-level anomaly detection. 2) iGAD for the first time discuss the anomalous attributes and substructures for the graph-level anomaly detection issue. iGAD enables the graph convolution to expose anomalous node attributes and the random walk kernel to identify anomalous substructures, which is not explored in previous work. In addition, we proposed a novel loss function to address the imbalanced labels.
>
> **Q2:** In Table 2: Why the authors do not give also the results for the Precision metric?
>
> **R2:** We list our considerations as follows: 1) Recall and Precision are a pair of negatively correlated indicators. F-score is a trade-off between Precision and Recall. When there are no specific application scenarios, an anomaly detection model that can achieve a high F-score is preferred. Hence, we choose F-score. 2) In experiments, the problem we face is detecting anticancer molecules. In this task, we lay particular emphasis on the recall of anti-cancer molecules. This is because the quantity of anti-cancer molecules is quite a few, we expect to recall as many anti-cancer molecules as possible. Therefore, we also choose Recall and especially Recall(A).
>
> **Q3:** In Table 2: Are all these results tested for statistical significance? It is important to mention the statistical test and score for these results, as the 20% of the datasets that is being tested, is a small dataset.
>
> **R3:** We did not carry out statistical significance tests. Table 2 reports the experiment results of all algorithms under 5-fold cross-validation. Specifically, a graph dataset will be divided five times, and almost all graphs have the opportunity to become testing data in a certain division. We run a model on five different data divisions separately and report the mean and standard deviation of model performance.
>
> Considering the reviewer’s concerns, we conducted 5-fold cross-validation experiments with more testing data to further verify iGAD’s effectiveness. Specifically, we conduct experiments on PC-3 and SW-620 (as shown in Table 1 of our submission, these two datasets with the most imbalanced data distribution), and the proportion of testing data varied from 30% to 60%. The experiment results demonstrate that the performance of iGAD keeps stable when reducing the quantity of training data (increasing the quantity of test data).
>
> |PC-3|20%|30%|35%|40%|45%|50%|60%|
> |---|---|---|---|---|---|---|---|
> |AUC|86.04+-1.14|85.49+-0.46|85.13+-0.73|85.33+-0.57|84.15+-0.88|84.75+-1.13|83.41+-1.05|
> |Recall|79.59+-0.41|79.82+-0.60|78.97+-0.58|78.98+-0.55|78.11+-0.62|78.33+-0.95|77.23+-1.12|
> |Recall(A)|73.69+-1.64|72.89+-2.77|71.15+-1.85|71.58+-0.99|70.20+-0.53|71.20+-2.02|69.95+-2.99|
> |F-score|63.50+-0.73|64.68+-1.24|64.43+-2.09|64.01+-1.25|63.34+-1.29|62.97+-0.80|61.86+-1.69|
>
> |SW-620|20%|30%|35%|40%|45%|50%|60%|
> |---|---|---|---|---|---|---|---|
> |AUC|85.82+-0.69|86.48+-1.02|86.25+-0.35|85.47+-0.74|84.59+-0.76|84.85+-0.49|83.18+-0.83|
> |Recall|79.64+-0.83|79.76+-0.78|79.42+-0.45|78.85+-0.60|78.42+-0.47|78.55+-0.34|76.94+-0.97|
> |Recall(A)|74.27+-2.99|74.05+-1.76|74.62+-1.50|72.12+-0.84|70.21+-2.13|71.39+-1.65|69.87+-2.99|
> |F-score|63.68+-1.56|64.04+-1.20|62.96+-1.12|63.74+-1.16|64.38+-0.88|63.70+-0.66|64.01+-1.09|
>
> **Q4:** Did the authors try bigger size of random walks (> 8 nodes)? To better understand this experiment?
>
> **R4:** Yes, please find the result in the anonymous link
> https://github.com/iGADrebuttal/9811_NeurIPS_rebuttal
>
> We observed a similar phenomenon as before. Increasing the length of random walks can improve the performance of iGAD. However, the performance improvement and the increase of the random walk length are not strictly monotonic.
>
> **Q5:** It would be useful to add in the Table 1 with the dataset statistics: the number of connected components in each graph and the avg. min. max. size of the connected components in each graph.
>
> **R5:** All graphs MCF-7 and PC-3 are fully connected. In MOLT-4, only one graph has an isolation node and all other graphs are fully connected. SW-620 is the same as MOLT-4. We will add the information to appendix.

---

> > ### Comment · Reviewer_qTND · 2022-08-08
> > **I acknowledge I have read all reviews and authors' response to them**
> >
> > I would like to thank the authors for addressing my comments, I have also checked the improved figures and the additional results in the provided GitHub link. I still believe that running statistical significance tests will make a stronger statement for the overall performance of the proposed model and the comparison methods.

---

> > > ### Author Response · Authors · 2022-08-09
> > > **Response to Follow Up Comments of Reviewer qTND**
> > >
> > > Thank you very much for your time and confirmation on our improved figures and additional results. We are glad that our response addressed your concerns.
> > >
> > > We are very grateful for your additional comment. Here, we report the relevant statistical significance tests on datasets MCF-7 and PC-3 in the following tables. Just as we expect, all p-values are less than 0.05, which verifies that the performance of iGAD is statistically significant compared with ten baseline algorithms.
> > >
> > > **The statistical significance result on MCF-7:**
> > > _______________________________________________________________________________________________________________
> > > |MCF-7|iGAD|iGAD|iGAD|iGAD|iGAD|iGAD|iGAD|iGAD|iGAD|iGAD|
> > > |---|---|---|---|---|---|---|---|---|---|---|
> > > ||**GCN**|**GCN-GAD**|**GIN**|**GIN-GAD**|**DGCNN**|**DGCNN-GAD**|**SOPOOL**|**SOPOOL-GAD**|**RWGNN**|**RWGNN-GAD**|
> > > |**AUC**|3.45e-11|2.60e-06|0.0004|7.19e-05|1.93e-05|3.46e-05|5.09e-06|0.72e-05|5.09e-06|3.59e-08|
> > > |**Recall**|4.24e-09|5.96e-05|0.0001|0.009|5.54e-08|0.0004|3.37e-08|0.0002|3.384e-08|5.51e-05|
> > > |**Recall(A)**|6.43e-12|5.96e-05|1.00e-05|0.002|1.38e-10|0.0001|2.51e-10|0.004|2.51e-10|9.31e-08|
> > > |**F-score**|2.71e-08|3.34e-06|0.01|0.0001|0.0001|0.0001|1.74e-05|0.003|2.51e-10|0.001|
> > >
> > > **The statistical significance test result on PC-3:**
> > > ______________________________________________________________________________________________________________
> > > |PC-3|iGAD|iGAD|iGAD|iGAD|iGAD|iGAD|iGAD|iGAD|iGAD|iGAD|
> > > |---|---|---|---|---|---|---|---|---|---|---|
> > > ||**GCN**|**GCN-GAD**|**GIN**|**GIN-GAD**|**DGCNN**|**DGCNN-GAD**|**SOPOOL**|**SOPOOL-GAD**|**RWGNN**|**RWGNN-GAD**|
> > > |**AUC**|1.95e-05|1.07e-06|0.0001|0.001|0.0003|6.27e-05|0.0016|1.80e-05|3.59e-06|8.40e-06|
> > > |**Recall**|1.12e-11|3.17e-06|4.53e-05|0.0005|6.59e-12|2.54e-05|0.0009|2.81e-06|8.12e-10|1.53e-06|
> > > |**Recall(A)**|3.007e-12|3.17e-06|4.87e-05|0.0026|1.98e-12|1.0003|0.0007|8.113e-05|8.393-11|0.0016|
> > > |**F-score**|2.16e-09|6.06e-09|0.0008|0.01|2.21e-10|1.49e-06|0.002|0.006|1.74e-06|1.83e-09|
> > > _______________________________________________________________________________________________________________
> > >
> > > As the discussion period between the review and author is close to the end, we do not have enough time to report the results on the other two datasets. We will add the complete statistical analysis results on all datasets in the final version of the paper.
> > >
> > > Thanks again for the additional comments, which indeed improve our paper! And we hope our response can make a stronger statement to align with your expectation this time.

---

> > > > ### Comment · Reviewer_qTND · 2022-08-09
> > > > **Thanks for providing the p-values**
> > > >
> > > > Thank you for providing the results, they look good! Since there is no space in the paper, I would suggest to add a comment in captions and in the discussion of the results that they were tested for statistical significance and that in all cases p-value < 0.05.

---

> > > > > ### Author Response · Authors · 2022-08-09
> > > > > **Re: Thanks for providing the p-values**
> > > > >
> > > > > Thanks so much for your bright idea. We will follow your suggestion to make a note of our statistical significance results in the captions.

---

### Meta-Review · Area_Chair_j8k1 · 2022-08-25

**Recommendation:** Accept
**Confidence:** Certain

**Metareview:**

This paper proposes a dual-discriminative graph neural network for detecting anomalous graphs given a set of graphs. For imbalanced data, a point wise mutual information (PMI) loss is used. The extensive experimental results using real-world datasets demonstrate the effectiveness of the proposed method. The graph-level anomaly detection is an important problem, and the motivation and its challenges are well presented. This paper is well-written. The proposed method of considering anomalous node attributes and anomalous graph structures is interesting.

**Award:**

No

---

### Decision · Program_Chairs · 2022-09-14

Accept